# MENTISOCULI: Revealing the Limits of Reasoning with Mental Imagery

**Jana Zeller** [1 2 3]  **Thaddäus Wiedemer** [1 3 4]  **Fanfei Li** [1 3]  **Thomas Klein** [1 3 4]  **Prasanna Mayilvahanan** [1 3 4]
**Matthias Bethge** [4]  **Felix Wichmann** [4]  **Ryan Cotterell** [2]  **Wieland Brendel** [1 3 4]

## Abstract

Frontier models are transitioning from *multimodal large language models* (MLLMs) that merely ingest visual information to *unified multimodal models* (UMMs) capable of native interleaved generation. This shift has sparked interest in using intermediate visualizations as a reasoning aid, akin to human *mental imagery*. Central to this idea is the ability to form, maintain, and manipulate visual representations in a goal-oriented manner. To evaluate and probe this capability, we develop MENTISOCULI, a procedural, stratified suite of multi-step reasoning problems amenable to visual solution, tuned to challenge frontier models. Evaluating visual strategies ranging from latent tokens to explicit generated imagery, we find they generally fail to improve performance. Analysis of UMMs specifically exposes a critical limitation: While they possess the textual reasoning capacity to solve a task and can sometimes generate correct visuals, they suffer from compounding generation errors and fail to leverage even ground-truth visualizations. Our findings suggest that despite their inherent appeal, *visual thoughts do not yet benefit model reasoning*. MENTISOCULI establishes the necessary foundation to analyze and close this gap across diverse model families.

## 1. Introduction

> *Words [...] do not seem to play any role in my mechanism of thought. The psychical entities which seem to serve as elements in thought are certain signs and more or less clear images.*
> – Albert Einstein (*Hadamard, 1954*)

Recent vision–language models (VLMs), including *multimodal large language models* (MLLMs), typically treat vision as a passive, input-only modality. However, we are now witnessing a shift towards *unified multimodal models* (UMMs) capable of native, interleaved generation. Frontier models like Emu3.5, Gemini 2.5 / 3 and many others are trained to not only perceive but also actively generate text, images, video, and audio (e.g., Cui et al., 2025; Google DeepMind, 2025b;d;c; Deng et al., 2025; Liu et al., 2025; Qu et al., 2025; Team, 2024; Xie et al., 2025; Chen et al., 2025).

With more capable multimodal models comes a growing awareness that complex reasoning tasks need not be tackled in language alone (Mi et al., 2025; Zheng et al., 2025; Fan et al., 2025; Chern et al., 2025; Hao et al., 2025; Tong et al., 2025; Liang et al., 2025). The premise is that dense visuals, spatial information, physical interaction, or object dynamics—in short, the complexities of real-world environments—are intrinsically difficult to textualize and may be better handled visually (Yang et al., 2024).

From an anthropocentric perspective, this is plausible: Our thinking inherently involves *mental imagery*—quasi-sensory experiences we can *observe* and, crucially, *manipulate* in the absence of external stimuli (Richardson, 1969). For example, designing a dress entails visualizing its different panels and making adjustments based solely on imagined observations of their composition. This capacity is not only *reproductive* but *constructive*; mental imagery is believed to play an important role in problem-solving and has been linked to the generation of new knowledge (Nanay, 2023).

Whether models support an analogous form of visual reasoning is an active field of study, with approaches spanning a spectrum of explicitness: On the *implicit* end, McCarty & Morales (2025) suggest that LLMs can solve pictorial tasks using only internal representations, though others argue that these mental visualizations are fragile (Sepehri et al., 2025). Moving toward *explicit* imagery, interleaved visual aids ranging from latent visual tokens (e.g., Yang et al., 2025) to generated images in UMMs (Zhou et al., 2025; Li et al., 2025a) find some success—though performance gains are inconsistent, especially in multi-step settings (Li et al., 2025b). Finally, on the *natively visual* end of the spectrum, Wiedemer et al. (2025) show that image editing models

[1]Max-Planck-Institute for Intelligent Systems, Tübingen, Germany [2]ETH Zurich, Zürich, Switzerland [3]ELLIS Institute Tübingen, Tübingen, Germany [4]University of Tübingen, Tübingen, Germany. Correspondence to: Jana Zeller <jana.zeller@tuebingen.mpg.de>.

*Proceedings of the 43rd International Conference on Machine Learning*, Seoul, South Korea. PMLR 306, 2026. Copyright 2026 by the author(s).

and video models can solve some reasoning tasks entirely visually, directly modifying pixels of the input image.

Overall, the utility of *machine mental imagery* is unclear. While the *capacity* for multimodal generation exists, attempts to leverage it for reasoning yield ambiguous results. Crucially, it remains unclear whether failures stem from fundamental reasoning deficits, flawed image generation, or an inability to interpret self-generated cues—and the field lacks a rigorous framework to disentangle these factors across different modalities.

We propose MENTISOCULI[1] to comprehensively study frontier models' ability to form, maintain, and repeatedly manipulate visual representations in a goal-oriented manner. MENTISOCULI consists of five multi-step visual reasoning tasks designed to be difficult to textualize yet intuitive for humans to solve visually. All tasks are procedurally generated across stratified difficulty levels. This design yields ground-truth visual chain-of-thought solutions for granular analysis and allows us to calibrate complexity while ensuring the benchmark's longevity through future extensions.

Benchmarking state-of-the-art MLLMs, UMMs, a latent reasoning model, and a generative video model, we find that explicit visual thoughts are currently ineffective; no visual intervention reliably outperforms text-only baselines. Further analysis of UMMs exposes a critical issue: Models often possess the *textual* reasoning capacity to solve a task and the *generative* capacity to (at least sometimes) create correct visualizations. However, they fail to integrate these skills—suffering from compounding generation errors over multiple steps and, surprisingly, even failing to leverage ground-truth visual aids. Our results suggest that despite the intuition behind mental imagery, architectures cannot yet bridge the gap between generation and reasoning.

---

In summary, we provide

1. MENTISOCULI: A procedural, stratified benchmark for multi-step reasoning with mental imagery, designed to challenge frontier models (Section 2).

2. An analysis of the spectrum of machine mental imagery, covering MLLMs, latent reasoning, UMMs, and video models (Sections 4.1 and 4.2).

3. Evidence that the failure of UMM visual reasoning stems from an inability to maintain consistency and to leverage visual aids (Section 4.3).

4. Human reference data, highlighting different reasoning budget allocation in humans and frontier models (Section 4.5).

---

[1]Latin for *eyes of the mind*, the concept of which goes back at least to Cicero (-55)

## 2. Designing MENTISOCULI

The term *visual reasoning* is used across a myriad of benchmarks targeting VLMs and MLLMs, yet remains ambiguous: The vast majority of existing benchmarks do not consider *reasoning visually*, but instead evaluate *reasoning about visual information* (Xu et al., 2025; Hao et al., 2025; Lyu et al., 2025). Instead, we aim to benchmark models' ability to *reason with mental imagery*: to use a more or less explicit, self-maintained visual representation space that can be modified at will to aid in reasoning.

Humans have a strong intuition about which problems are naturally solved by forming and manipulating mental images. Because this intuition is difficult to formalize directly, we list a set of desiderata designed to make the intermediate steps naturally visual and to prevent textual or symbolic shortcuts:

1. **Visual nature**    Tasks should test understanding of spatial relations, geometric constraints, or object transformations, rather than common knowledge or mere symbolic logic. While visual reasoning might aid abstract or mathematical reasoning, visualizations that are not grounded in the problem statement are hard to verify and evaluate.

2. **High information density**    To be inefficient to solve via pure text, tasks should avoid grid-worlds or symbolic arrangements that are easily transcribed to short text descriptions (e.g., "Piece A is at (0, 1)"). Instead, tasks should involve complex shapes, continuous and off-grid transformations, or fine-grained visual details.

3. **Sequential manipulation**    To evaluate a model's ability to *maintain* a consistent visual state over time, tasks should require repeated updates to mental imagery, and actions should depend on the outcomes of previous manipulations. Solution sequences should be discrete to enable evaluation of models limited to image generation.

4. **Procedural**    Tasks should be easy to generate, including a ground-truth solution with visualizations, enabling deeper analysis. Additionally, procedural generation provides a mechanism to address data contamination in the future, ensuring the benchmark's longevity.

5. **Stratified**    Tasks should have a clear knob to control complexity (e.g., number of steps or objects). This allows us to identify the breaking point of frontier models and maintain the benchmark into the future by releasing higher-complexity problem instances.

6. **Generative feasibility**    Current model constraints should be respected. This includes visual states that are representable in 2D projections (i.e., not involving ambiguous depth cues or occlusions) and details that remain legible at standard resolutions.

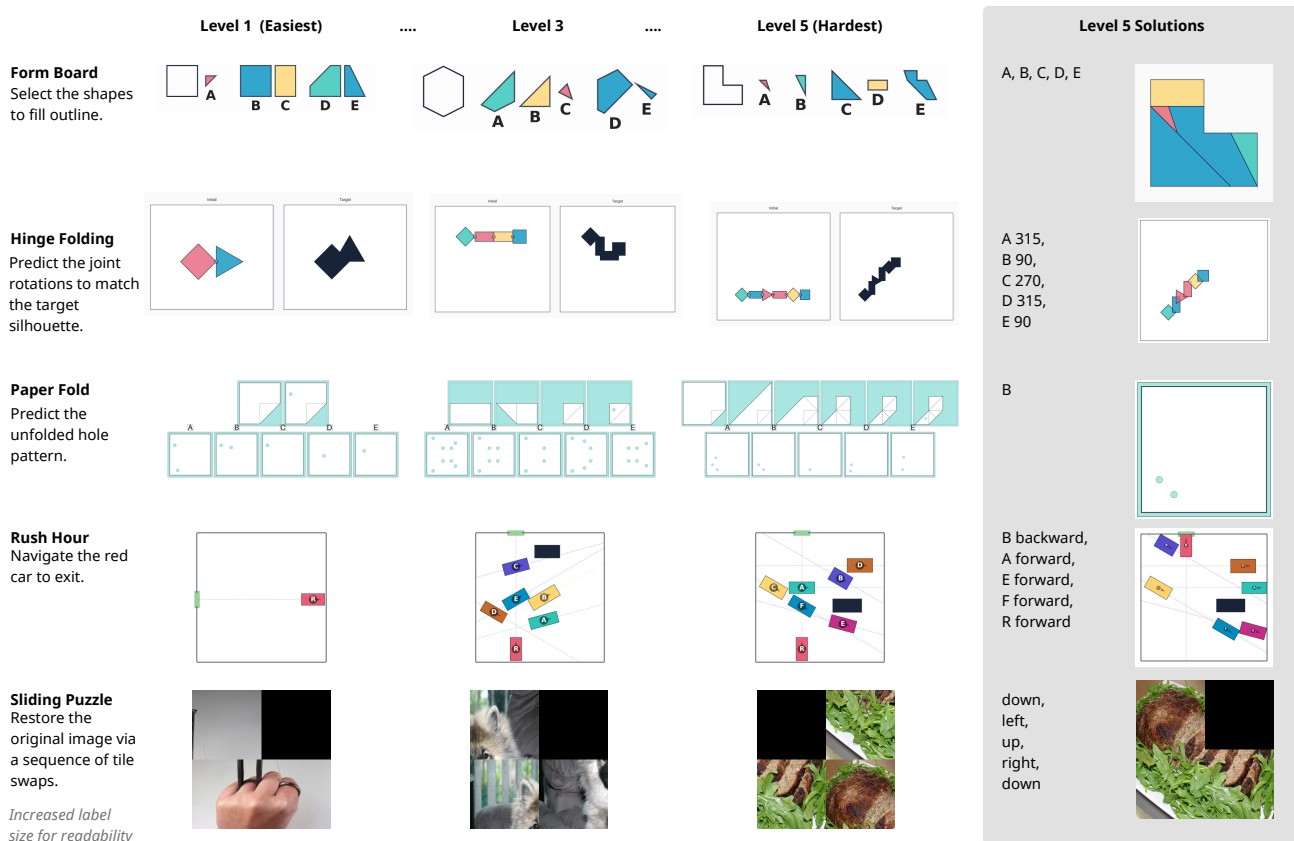

*Figure 1.* **MENTISOCULI comprises five visual reasoning tasks designed to be best-solved with mental imagery**. Collectively, the tasks require models to solve multi-step reasoning problems with geometric constraints. Success hinges on the ability to maintain a visual representation with high fidelity and consistent geometry under affine transformations. Each task is procedurally generated across five difficulty levels, scaling with the number of operations required from one (left) to five (right); see Appendix A for details.

While several benchmarks examine reasoning with interleaved images, they frequently fall short of our desiderata: Zebra-CoT and MIRA violate the visual nature requirement by relying on prior knowledge (Li et al., 2025a; Zhou et al., 2025). STARE and similar benchmarks (Li et al., 2025b; Hao et al., 2025; Wu et al., 2024; Chollet, 2019; Ramakrishnan et al., 2024) exhibit low information density, utilizing grid-based layouts that are trivially transcribed. Many tasks proposed by Chollet (2019); Lyu et al. (2025); Ramakrishnan et al. (2024); Huang et al. (2025); Sepehri et al. (2025) lack sequential manipulation, requiring only a single rule application or fill-in-the-blank completion. Further, several benchmarks are not strictly procedural due to manual crafting or a lack of generation code (e.g., MIRA), or suffer from limited sample variety and a lack of stratified difficulty levels (e.g., STARE). Finally, Artificial Phantasia proposes a purely linguistic task to measure mental imagery (McCarty & Morales, 2025). While individual tasks in prior work occasionally satisfy our criteria (e.g., in VisFactor, Huang et al., 2025), MENTISOCULI is the first benchmark exclusively dedicated to this rigorous category of mental imagery.

We release MENTISOCULI with the following procedural tasks, each at five levels of difficulty (see Figure 1).

**FORM BOARD** Derived from Ekstrom & Harman (1976), this task probes the ability to *compare shapes*, *understand spatial constraints*, and *maintain geometry* under translation. Models must identify the subset of candidate shapes that cover the target silhouette without gaps or overlaps. Our implementation builds on Huang et al. (2025).

**HINGE FOLDING** This task retains the need to *compare shapes* and *maintain geometry*, but introduces the complexity of *mental rotation* and *object dependencies*. Models must predict the discrete rotation angle (in 90° steps) for each hinge in a chain of polygons to form a target silhouette.

**PAPER FOLD** Adapted from Ekstrom & Harman (1976) and Huang et al. (2025), this task requires maintaining spatial locations under *reflection symmetry*, demanding higher *spatial fidelity* than previous tasks. Given an image showing a sequence of folds and a hole punch applied to a paper sheet, models must identify the correct unfolded pattern.

RUSH HOUR   This task tests *multi-step planning* under *dynamic geometric constraints*. Models must navigate the red vehicle out of a crowded lot by moving blocking vehicles. To prevent symbolic grid-based shortcuts, vehicles are not axis-aligned and have continuous-valued positions, though actions are discrete `forward`/`backward` commands.

SLIDING PUZZLE   This task evaluates *multi-step planning* with a focus on *visual coherence*. The pieces of a natural image are permuted on a grid, with one piece missing. Models must output the sequence of moves (`up, down, left, right`) of the empty tile to restore the image.

We control the difficulty of each task via the minimum number of steps (moves, folds, etc.) required to reach the solution. We generate 30 samples per level for the initial version of the benchmark; see Appendix A for details on the generators. As we will show, Level 5 is more than sufficient to challenge current models. We release our code to generate more challenging problem instances in the future.

## 3. Evaluation

### 3.1. Model families

We compare the following model families, spanning a spectrum from implicit to explicit visual reasoning. Prompts and hyperparameters are detailed in Appendices G and I. We query models up to three times to obtain an answer and use the highest reasoning budget unless specified otherwise.

- **Multimodal large language models (MLLMs)** on the *implicit* end of the spectrum produce text-only outputs and can only expose or interleave visual representations through textual visual proxies such as ASCII art. We query **Gemini 2.5** (Flash), **Gemini 3** (Pro), **GPT-5.1**, and **Qwen3-VL** (235B-A22B Thinking) (Google DeepMind, 2025b;d; OpenAI, 2026; Team, 2025).

- **Latent visual reasoning models** produce text reasoning chains interleaved with visually-grounded latents. This category lacks widely-established models; we fine-tune Qwen2.5-VL-32B (Bai et al., 2025) on RUSH HOUR using the **Mirage** and LatentSketchpad (**LS**) frameworks (Yang et al., 2025; Zhang et al., 2025), which were both explicitly designed for visual reasoning.

- **Unified multimodal models (UMMs)** can *explicitly* visualize states as images interleaved into the reasoning chain. We specifically prompt for and only evaluate samples with generated visualizations (see Appendix G.1). In this category, we query **Gemini 2.5-I** (Flash Image), **Gemini 3-I** (Pro Image) and **Emu 3.5** (Google DeepMind, 2025b;c; Cui et al., 2025).

- **Video models** represent the *natively visual* end of the spectrum, producing purely visual rollouts conditioned on a prompt and an initial frame. After comparing multiple video models (see Appendix F.2), we report results for **Veo 3.1** and **Wan 2.6** (Google DeepMind, 2025a; Alibaba Cloud, 2025).

### 3.2. Automated scoring

**Text outputs**   We evaluate **MLLMs**, **UMMs**, and **latent visual reasoning models** on the text output that they produce. For FORM BOARD and PAPER FOLD, we score answers as correct only if the model's predicted option(s) exactly match the ground-truth label(s). For HINGE FOLDING and SLIDING PUZZLE, we parse predicted action sequences and simulate them in the corresponding environment. Predictions are correct only if the simulated terminal state satisfies the task goal (target silhouette matched original image reconstructed). For RUSH HOUR in accordance with chance performance, we count any trace as valid that contains one correct final state. Outputs that reference invalid identifiers (e.g., non-existent vehicles) or contain invalid moves (e.g., out-of-bounds actions) are scored as incorrect.

**Visual outputs**   For RUSH HOUR, we follow Wiedemer et al. (2025) and implement an automatic rater for **video model** output (see Section 4.2). The rater processes videos frame-by-frame, using color and spatial consistency to recover object identities and trajectories. From the trajectories, we extract an implied sequence of actions via a lenient heuristic: We only consider each vehicle's first move and relative order of moves, ignoring minor visual artifacts (color changes, minor distortions, etc.) and continued motion after reaching the goal. However, large scene changes, including the introduction of spurious objects, immediately invalidate a sample. Analogous to the text-based validity checks, parsed actions are verified by simulation.

### 3.3. Human reference data

To gauge top human performance, we conduct a psychophysical experiment on RUSH HOUR, which resembles standard IQ tests. Thus, we can assume performance to be normally distributed among the general population. Since we only require an upper bound, we investigate a small population of PhD students ($n = 5$, 2f/3m, mean age 27), yielding high-quality data. The population includes two of the first authors who were familiar with the task, while other participants remained naive. Crucially, all humans were instructed to respond as quickly as possible, such that response time is a proxy for perceived difficulty. For a comprehensive description of the experimental setup, see Appendix B.

### 3.4. Chance Performance

For FORM BOARD and PAPER FOLD, we assume a uniform distribution over possible answers. For HINGE FOLD-

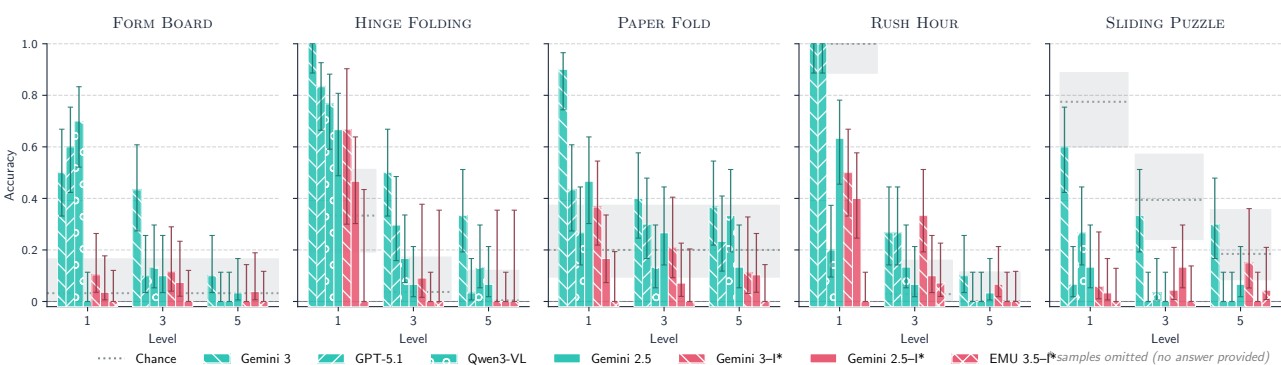

*Figure 2.* **MLLMs and UMMs display similar failure patterns across tasks**: Performance degrades noticeably with difficulty and falls below chance at Level 5, indicating that visual reasoning limitations are task-agnostic. Data for all levels in Figure 12.

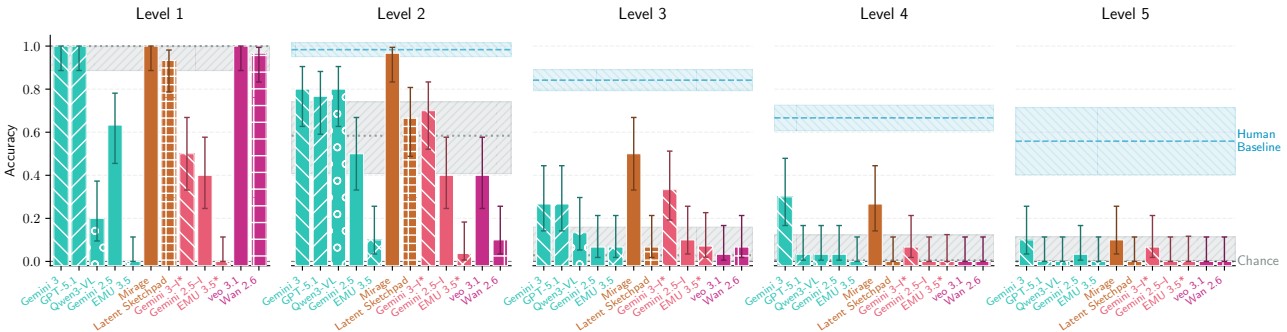

*Figure 3.* **Different kinds of mental imagery do not greatly improve multi-step reasoning on RUSH HOUR**: Compared to **MLLMs**, the **latent visual reasoning models** which are fine-tuned to generate interleaved visual latent tokens shows some improvement (especially considering its relatively weak base model), but with diminishing returns at harder levels. In contrast, **UMMs** that interleave generated images and texts generally perform below their MLLM counterparts. The **video models** operate purely in pixel space and break down quickly as difficulty increases. *\* samples omitted (no answer provided)*

ING, we additionally assume the model to trivially infer the correct number of steps, such that chance performance decreases over levels. For the planning tasks RUSH HOUR and SLIDING PUZZLE, we report the probability that a random six-step action sequence reaches the goal state at any point, accounting for (limited) backtracking.

## 4. Results

### 4.1. SotA multimodal model performance across tasks

We begin by benchmarking the most capable models—state-of-the-art **MLLMs** with text-only reasoning and **UMMs** with interleaved text and image reasoning traces—on all tasks, see Figure 2.

Across tasks, performance degrades noticeably as difficulty increases, validating our stratification. While accuracies vary, the relative ranking of models is largely consistent: **Gemini 3** performs best, followed by **GPT-5.1** and **Qwen3-VL**. Both **Gemini 3-I** and **Gemini 2.5-I** often lag behind their corresponding **MLLMs**—we analyze this more closely in Sections 4.2 and 4.3.

With the exception of **Gemini 3**, models fail to reliably exceed chance even at Level 1 on all tasks except FORM BOARD. Thus, performance is often limited already at the level of extracting a single valid action from the visual state, rather than by long-horizon reasoning. As difficulty increases, this weakness compounds: by Level 5, all models operate at or below chance. Notably, even cases of sub-chance performance arise. This is mainly caused by early termination and under-utilization of the action budget, not incorrect state transitions.

> **Takeaway 1** MENTISOCULI is far from saturated. Below-chance performance at Level 5 highlights the limitations of SotA visual reasoning models (Figure 2).

### 4.2. Comparing model families on RUSH HOUR

Given the stability of relative performance across tasks, we focus on a single representative task in Figure 3 to compare the full spectrum of reasoning paradigms, from implicit text-only reasoning in **MLLMs** to explicit visual generation in **UMMs**. We select RUSH HOUR because it enables a unified, action-based evaluation across all model families

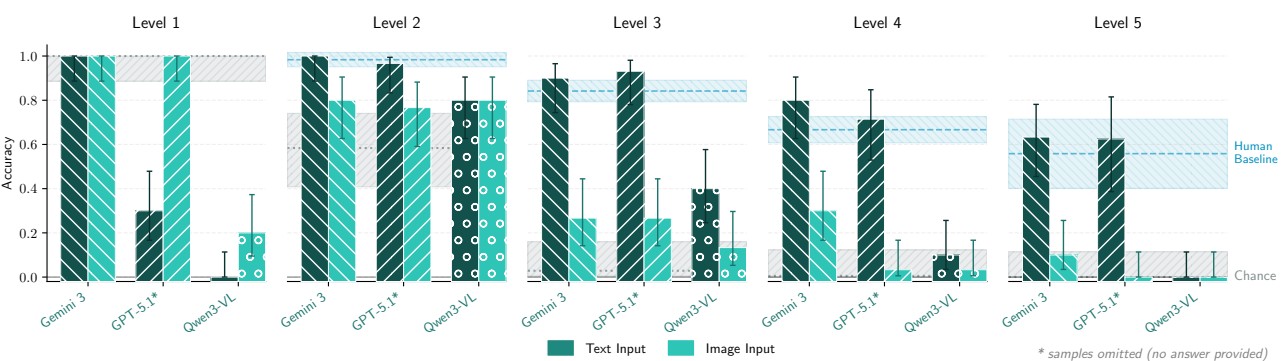

Figure 4. **MLLMs** **have the** *competence* **to solve** RUSH HOUR when prompted with a transcription of the task. **Gemini 3** and **GPT-5.1** even perform on par with humans, even though the text-only RUSH HOUR requires mathematically solving for possible collisions.

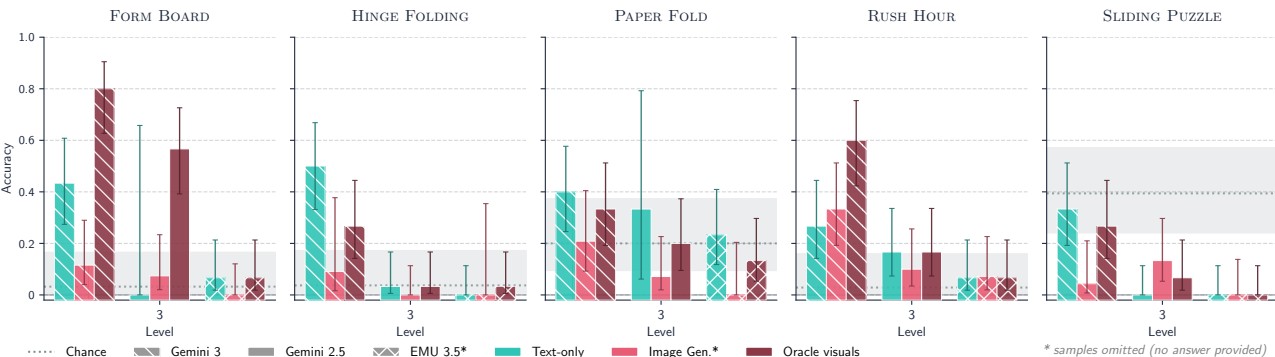

Figure 5. **UMM** **performance faces a dual issue**: *Generation errors* are ubiquitous—performance on all tasks increases with oracle visualizations. However, on most tasks, UMMs fail to utilize even correct visuals to aid their reasoning, which we term *interpretation errors*. Data for all levels in Figure 13.

while covering a broad range of difficulty levels, before analyzing failure modes across tasks in Section 4.3.

**MLLMs vs. latent reasoning** Fine-tuned on 200 samples per level, **Mirage** outperforms **MLLMs** on Levels 2–3, but this advantage is brittle: at higher levels it merely matches **Gemini 3** and drops to near-chance at Level 5. Compared to text-only fine-tuning we observe a limited effect of latent reasoning (see Appendix E). This suggests that latent visual tokens offer only limited gains, particularly for more actions.

**MLLMs vs. UMMs** Contrary to our intuition, we see no improvements moving from **Gemini 3** to **Gemini 3-I** or from **Gemini 2.5** to **Gemini 2.5-I**. In fact, text-only **MLLMs** frequently outperform **UMMs**. This implies that *explicit interleaved visualizations* currently provide no consistent benefit to *implicit* multimodal reasoning. A further analysis of this phenomenon follows in Section 4.3.

**Video models** Despite the lenient scoring policy (Section 3.2), **Veo 3.1** and **Wan 2.6** never exceed chance performance. Yet, **Veo 3.1**'s ability to match or exceed **Gemini 2.5-I** on lower levels lends credence to the potential for *natively visual* reasoning.

**Human-machine gap** While **Mirage** comes close on Level 2, models generally fall far behind human performance. Human performance is consistent between subjects and drops with higher difficulty (see Figures 3 and 10). A detailed analysis follows in Section 4.5.

> **Takeaway 2** Explicit visual thought is currently ineffective. We find no evidence that self-generated imagery (latent, interleaved, or video-based) improves text-only reasoning in multi-step visual problems (Figure 3).

### 4.3. What is holding MLLMs and UMMs back?

**Symbolic vs. sensory reasoning** While our tasks are designed to be non-isomorphic to *low-token* text, we can still provide a lossless (if complex) transcription of RUSH HOUR: We specify the parking lot size and the exit location, as well as each car's center coordinates, spatial extent, orientation, and admissible motion axis (see Appendix H.1). For humans, reasoning about the problem in this formulation is exceedingly cumbersome compared to eyeballing a visual solution. But it allows us to shift the reasoning problem away from visual understanding and planning with mental

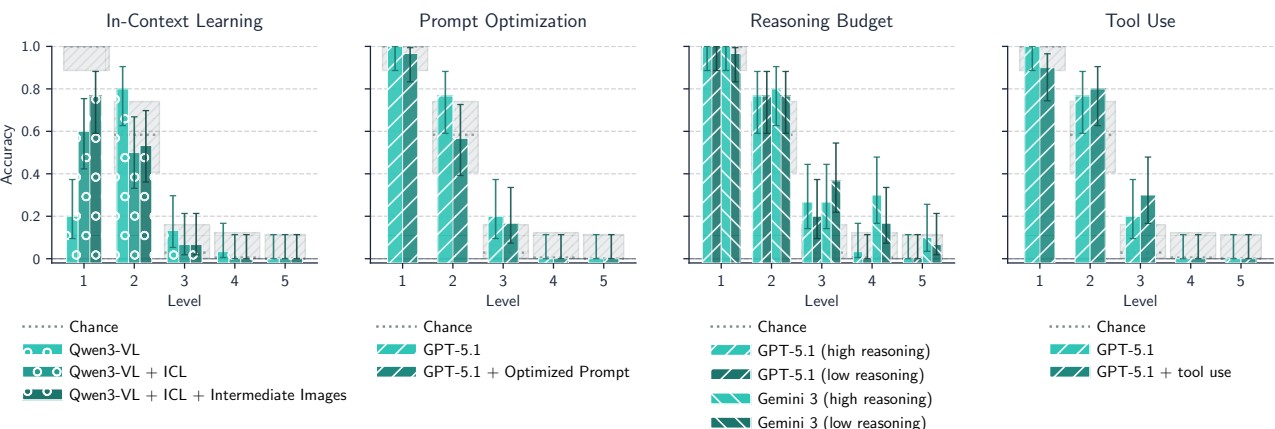

Figure 6. **Techniques that improve language-based reasoning fail to benefit visual reasoning**: In-context learning (ICL), prompt optimization, increased reasoning budget, and tool use yield no consistent gains, especially at higher levels. The tool use and prompt optimization experiments were conducted with low reasoning.

imagery to mathematically solving geometrical constraints.

The comparison in Figure 4 shows that the task is not inherently beyond the reach of MLLMs: Gemini 3 and GPT-5.1 possess the reasoning capabilities to solve RUSH HOUR, even in this (from a human perspective) complex form.

This makes visual understanding and manipulation the main bottleneck. UMMs possess linguistic abilities mirroring those of corresponding MLLMs. They also understand and generate visuals with high precision (see Figures 17 and 18). So why do they not outperform corresponding MLLMs?

**Reasoning with oracle visual chain-of-thought** Where does the failure of UMMs to reason with interleaved images stem from? Is it an inability to generate correct visuals (*generation error*)? Or do UMMs fail to utilize generated visuals to aid reasoning (*interpretation error*)? To test this, we replace self-generated imagery with oracle visuals (see Appendix H.2) in UMMs' chain-of-thought (CoT).

As illustrated in Figure 5, providing oracle visuals highlights a significant performance ceiling for UMMs. While oracles allow Gemini 3-I and Gemini 2.5-I to achieve peak accuracy on FORM BOARD—far exceeding chance and their underlying MLLM—the impact is more varied in other domains. Notably, in HINGE FOLDING and PAPER FOLD, oracle visuals only close the performance gap to the underlying MLLM. Emu 3.5 demonstrates similar trends but consistently trails the Gemini models across all conditions. This suggests that beyond the pervasive issue of *generation errors*, models face substantial *interpretation errors*.

Yet, on other tasks, oracle visuals are still not sufficient to reliably meet or exceed chance performance. Thus, UMMs also suffer from *interpretation errors* as they fail to interpret visual states as actionable evidence for decision-making.

**Coupling of Reasoning in Images and Text** If a UMM's image generation is faithful and its interpretation of those generated visuals is accurate, the actions it proposes in text should match the actions enacted in its generated image sequence. In order to assess this, we we adapt our video auto-rater for RUSH HOUR to extract moves from image-only outputs and compare, per puzzle, which modality solves it.

Figure 7 shows a consistent pattern across UMMs: the two channels do not just differ in accuracy, they solve largely *different* puzzles. Even on the easiest level, roughly half of all solved puzzles are solved by exactly one of the two channels. Which channel dominates flips with difficulty: at Level 1 image rollouts solve puzzles text reasoning misses, but from Level 2 onward the inversion is sharp — the text channel carries multi-step planning that the image channel cannot.

The joint generation+interpretation failure does not shrink with model capability. Gemini 3-I, with nearly twice Gemini 2.5-I's text accuracy, shows a higher fraction of disagreeing puzzles. The additional puzzles its text channel solves are not the ones its image channel was closest to solving, so a stronger underlying MLLM does not by itself reduce the coupling failure.

> **Takeaway 3** Frontier models possess the *competence* to solve RUSH HOUR (Figure 4). Yet, *performance* on visual tasks suffers from generation errors and a deeper inability to utilize visuals in planning (Figures 5 and 7).

### 4.4. Limits of common reasoning enhancements

Do techniques that improve language-based reasoning also elevate performance on visual reasoning tasks? We evaluate four such approaches on RUSH HOUR:

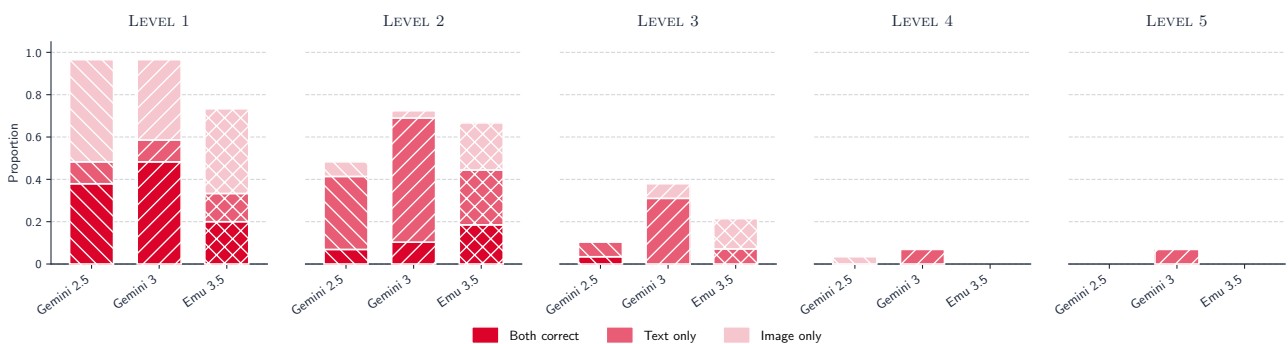

*Figure 7.* **Image- and text-derived actions are weakly coupled** Per-modality performance of three **UMM**s on RUSH HOUR. Text only and Image only are decoupled outcomes where exactly one modality solves the puzzle. Even on the easiest level, decoupled outcomes account for half of all puzzles.

**In-context learning**  Providing ICL examples yields no systematic improvement beyond Level 1. Moreover, we observe no difference between ICL examples that include images and those that do not (see Appendix I.7).

**Prompt optimization**  Optimizing prompts (57 variants over 50 iterations) using OpenEvolve (Sharma, 2025) does not improve performance over our default prompt. The optimized prompt can be found in Appendix I.8.

**Reasoning budget**  Increasing reasoning effort does not improve accuracy. Although **GPT-5.1** and **Gemini 3** use substantially more tokens under high reasoning settings (on average $13\times$ more for **GPT-5.1**), performance remains largely unchanged across difficulty levels. Results on all tasks can be found in Appendix D.2.

**Tool use**  Enabling tool use yields no meaningful gains. The model primarily uses image preprocessing tools (cropping, resizing) without improving downstream accuracy.

> **Takeaway 4**  Established techniques to improve text-based reasoning—including ICL, prompt optimization, increased reasoning budget, and tool use—fail to provide systematic gains for visual reasoning (Figure 6).

### 4.5. Comparing humans and machines

**Mapping performance to response time**  To contextualize the best model performance, we compare **Gemini 3** to time-constrained humans in Figure 8. We artificially reduce the available time to an arbitrary threshold $t$ by only considering trials with a correct response given in $< t$ seconds. Thus, we obtain time-constrained observers without re-testing with different limits. Evidently, humans are quite capable of solving the task, achieving more than 60% accuracy at Level 5 (Figure 10). **Gemini 3** then falls between humans limited to $5-10$s.

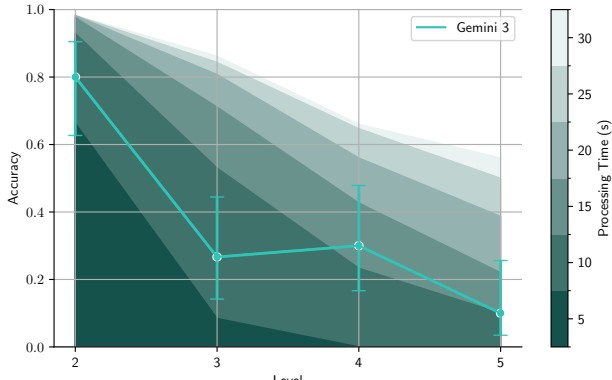

*Figure 8.* **Gemini 3** performs like humans at $5-10$s. We plot average human performance at each difficulty level, while simulating different thinking time cutoffs ($5-30$s).

**Human vs. machine adapative reasoning effort**  Beyond absolute performance, humans and models differ in how they allocate effort across difficulty levels. Humans reliably spend more time on higher-level puzzles, indicating a consistent internal difficulty assessment. In contrast, **Gemini 3** shows no increase in token usage from Level 3 to Level 5 (Figure 9). Unlike humans, it does not dynamically adjust its internal reasoning process in response to increasing complexity (see Appendix C for other models).

> **Takeaway 5**  Unlike humans, models don't increase their reasoning effort in response to visual–spatial complexity, meaning they lack adaptive reasoning depth (Figure 9).

## 5. Discussion & conclusion

**Is explicit visual thought a dead end?**  We currently don't observe UMMs or video models using self-generated mental imagery to outperform text-only reasoning (Section 4.2). Yet, our experiments also suggest that frontier models already possess the *competence* to solve our tasks (Figure 4).

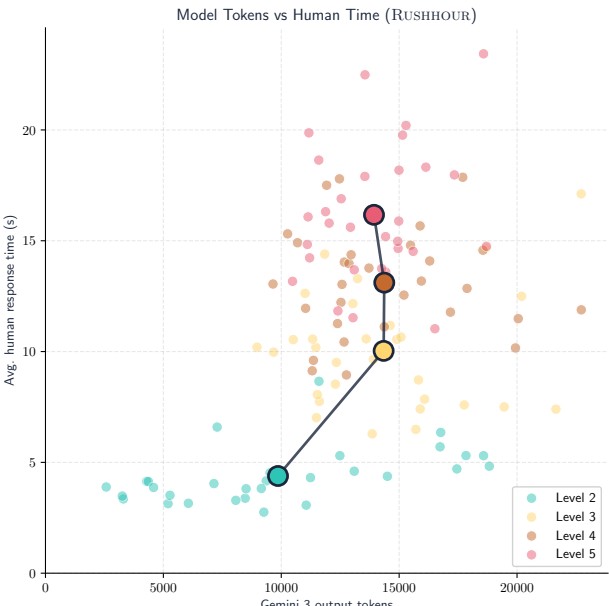

*Figure 9.* **Humans and machines allocate reasoning effort differently**: Humans spend more time on harder problems, but Gemini 3 does not use more tokens. More models in Appendix C.

We also see that performance could increase on some tasks if *generation errors* were curbed (Figure 5), and we can speculate that fixing *interpretation errors* might yield further gains. Looking at related literature (e.g., Mayilvahanan et al., 2025), it is likely that getting models to ground their decisions in mental imagery will require dedicated training data and a greater focus on multi-step visual reasoning by model developers.

**The fragility of visual thought** Our observation that models often fail to benefit from ground-truth visual chains of thought (see Figure 5) affirms prior work. For example, Li et al. (2025b) report variable effects of visual traces, while Zhou et al. (2025) observe average improvements that obscure task-level heterogeneity. We observe a similar pattern: Visual aids can be helpful in some settings, but their effectiveness is neither uniform nor reliable. This suggests that the key question is not whether mental imagery is beneficial in general, but which visual aids are useful for which tasks.

**The high price of visualization** Beyond accuracy, it remains to be seen whether machine mental imagery can become economically viable. Generating a video reasoning trace with Veo 3.1 costs \$3.2 per sample—over 21× more than Gemini 2.5-I, and over 60,000× more than Gemini 2.5—despite all three approaches yielding roughly similar performance. For UMMs or video models to replace text-centric MLLMs on specific tasks, they would have to justify this overhead through clear performance gains or qualitatively new capabilities.

**Conclusion** In this work, we consider visual reasoning *with* imagery rather than merely *about* images. Our results show that current models struggle to effectively use visual aids as actionable evidence, even when correct visualizations are given. MENTISOCULI provides a controlled, procedural testbed for isolating this failure mode and distinguishing reasoning capacity from representational alignment. We view MENTISOCULI as a step toward understanding when and why (explicit) visual representations support reasoning, and toward clearer criteria for progress in machine mental imagery.

## Conflict of Interest Disclosure

TW was employed by Google DeepMind during a part of this project, serving in an advisory capacity during that time. This work used Google Cloud Platform credits provided through Google's Gemini Academic Program Award. There are no further assosiactions with any of the tested models.

## Contribution Statement

The project was led by JZ and TW. The initial idea was pitched by TW and refined with the help of JZ, PM, WB, TK, and RC. The tasks were designed and implemented by JZ with inputs from TW, PM, TK, WB, MB. FL implemented the video auto-rater and tested different video models. All other training, inference, and evaluation were run by JZ, with help on the evaluation design by TW, RC, PM, and WB. Human experiments were designed and analyzed by TK with input from FW and conducted by JZ. The manuscript was written by JZ and TW, with help from FL for sections on the video models and from TK for sections on human experiments, and general comments from RC and WB.

## Impact Statement

This paper presents work aimed at advancing the evaluation of multimodal reasoning systems. We introduce a benchmark to better understand the limits of current models in reasoning with visual representations and to support more rigorous analysis of their capabilities and failure modes. We do not anticipate significant negative societal impacts arising directly from this work. As with most advances in machine learning research, there may be broader downstream applications, but we believe these are well understood and do not warrant specific discussion here.

## Acknowledgements

We would like to thank Robert Geirhos and Jack Brady for helpful discussions. We would also like to thank all our participants for taking part in our experiments.

Funded, in part, by the Collaborative Research Centre (CRC) "Robust Vision – Inference Principles and Neural Mechanisms" of the German Research Foundation (DFG; SFB 1233), project number 276693517. FAW acknowledges funding by the BBVA Foundation Programme Grant "Harnessing Vision Science to Overcome the Critical Limitations of Artificial Neural Networks". This work was additionally supported by the German Federal Ministry of Education and Research (BMBF): Tübingen AI Center, FKZ: 01IS18039A. WB acknowledges financial support via an Emmy Noether Grant funded by the German Research Foundation (DFG) under grant no. BR 6382/1-1 and via the Open Philanthropy Foundation funded by the Good Ventures Foundation. WB, FAW, and MB are members of the Machine Learning Cluster of Excellence, EXC number 2064/1 – Project number 390727645. Authors acknowledge funding by the Federal Ministry of Research, Technology and Space of Germany (BMFTR, formerly BMBF) under grant no. 01IS24085C (OPENHAFM). The authors thank the International Max Planck Research School for Intelligent Systems (IMPRS-IS) for supporting TK, TW, FL, and PM. JZ is supported by the Max Planck ETH Center for Learning Systems.

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

# A. Automatic Puzzle Generation

To construct MENTISOCULI, we implement five task-specific auto-generators that can produce infinitely many puzzle instances with controllable difficulty. For each instance, the generator produces a single question image specifying the full problem state and a ground-truth visual chain of thought capturing the sequence of intermediate states required to reach the solution. Restricting task input to a single image ensures that the same instance format can be used across models that allow for multiple image inputs (UMMs, MLLMs, latent visual reasoning models) and models that only allow one initial image input (video models), and our human study.

**FORM BOARD** We build on the FORM BOARD implementation of Huang et al. (2025). Each instance is generated by cutting an initial shape into a set of target pieces that together form the ground-truth solution. To avoid trivial matching, all target pieces are constrained to be pairwise distinct. The distractor pieces are generated by subdividing the target pieces so that their areas differ sufficiently from all the correct solution pieces, this ensures that the false candidate shapes are indeed false. The visual CoT depicts the progressive reconstruction of the target shape, adding one piece at a time in the correct location.

**HINGE FOLDING** Instances consist of rigid shapes connected by hinges. We generate either chains of identical shapes or chains with varying shapes. Hinge rotation angles are sampled in $90°$ increments. Additionally, a minimum area has to be visible in the folded configuration. The visual CoT shows the sequential application of hinge rotations corresponding to each folding step.

**PAPER FOLD** We build on the generator of Huang et al. (2025). Each instance is created by randomly sampling a sequence of folds and a hole-punch location, while tracking the resulting hole pattern through the folding process. Negative answer options are generated by sampling hole configurations that are globally similar, i.e. similar amount and placement of holes, but guaranteed to differ by at least one hole beyond a fixed minimum spatial threshold. We additionally generate a visual CoT that explicitly visualizes the unfolding process step by step. Both the minimal-difference constraint between answer options and the generation of the unfolding visual CoT are novel relative to prior work.

**RUSH HOUR** We generate RUSH HOUR instances by first sampling an exit location and placing the red car on the opposite side of the board. Depending on the difficulty level, we then sample zero to two primary blocking cars aligned with the red car's movement axis. Additional cars and obstacles are placed randomly, with a bias toward blocking the movement of these primary blockers to induce secondary dependencies. Each instance is solved using breadth-first search to ensure solvability and minimal solution length. To avoid visually ambiguous near-collisions, we re-evaluate the solution using slightly enlarged car sizes and discard instances where the red car no longer reaches the exit.

**SLIDING PUZZLE** We build on the SLIDING PUZZLE generator of de Oliveira et al. (2024). Each instance is constructed by sampling an image from ImageNet-1k (Deng et al., 2009), randomly selecting a tile to replace with the blank tile, and applying a sequence of valid moves to scramble the puzzle. This avoids permutation parity constraints and produces reachable states. To ensure correct difficulty classification and a minimal visual CoT, we subsequently solve the scrambled puzzle and record the shortest solution trajectory. Unlike the original implementation, the blank tile is sampled at arbitrary positions rather than being fixed in the bottom-right corner, which would otherwise enable shortcut strategies. All difficulty levels share the same underlying images.

# B. Psychophysics Experiment

To collect human reference data, we implement a minimalist web application, which displays a puzzle instance and lets participants respond via keyboard input. Experiments were conducted in a quiet environment on standard MacBook screens. We employ a block design, where 10 instances are sequentially presented for a maximum of 30 seconds each, with breaks of arbitrary length between blocks, so that participants can rest their eyes. At the beginning of each experiment, instructions (see Appendix I.4) are presented that closely resemble those given to the models. Then, seven practice trials are presented, which only serve the purpose of familiarizing participants with the interface. For the first two practice trials, we show the correct response, so that participants can learn the correct response format. Each car in the RUSH HOUR instance has a letter and can go forward or backward, so to indicate that car A should go forward and car B should go backward, participants would respond *AFBB*. Each block contains trials of varying difficulty, thus keeping the difficulty of blocks balanced, which

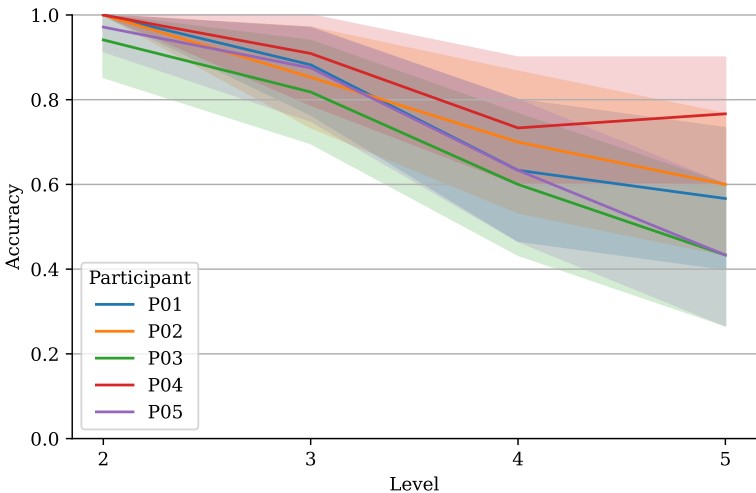

*Figure 10.* **Human subjects perform similarly** We plot the difficulty levels against performance for all our human subjects. Evidently, differences between humans present themselves only at the hardest difficulty level. Overall, our subjects perform similarly and, crucially, on par with the authors, demonstrating that we successfully investigated subjects close to the performance ceiling.

aids in motivation. We provide positive and negative feedback after a response was given, but reveal the default solution only during the practice phase. All participants gave informed consent, and we obtained IRB approval for the human study.

The results validate our experiment design: The performance of tested humans is closely aligned, even though two of them are authors and thus intimately familiar with the tasks (see Figure 10). Therefore, we are confident that our small-$N$ design is adequate and sufficient for the purposes of this study (see also Smith & Little, 2018).

## C. Reasoning Budget Correlation on More Models

Our analysis reveals that access to visual reasoning traces effectively aligns model compute with human difficulty metrics. As shown in Figure 11, providing models with visual CoT (i.e., **Gemini 2.5-I** with oracle visual CoT and **Qwen3-VL** with in-context-learning examples containing visual CoT) induces a highly linear relationship ($R^2 \geq 0.98$) between the number of tokens spent on a problem and the time the average human requires to solve it. This suggests that these visual CoT-guided models mirror human cognitive scaling when navigating the puzzle's state space.

However, we find that such alignment is not a definitive predictor of task success. While the oracle-guided models are the most aligned, **Gemini 3** demonstrates superior downstream performance—particularly on the most challenging Level 5 puzzles—despite having a lower alignment score ($R^2 = 0.68$) also compared to other **MLLMs**.

## D. Results Across All Difficulties & Tasks

### D.1. Performance Across All Difficulty Levels

Figures 12 and 13 report results for all five difficulty levels of each task. We observe a smooth and largely monotonic accuracy degradation as difficulty increases, with no qualitative regime changes between adjacent levels. Crucially, the relative ordering of model families and the effect of interleaved visual reasoning are stable across levels. Interleaved visual chains of thought yield consistent but task-specific effects across difficulty, benefiting static spatial tasks while providing little to no improvement for planning-dominated tasks.

Additionally, we test whether the visual Chain-of-thought experiments are hold back by the fact that the CoT is appended to the prompt, therefore increasing the multi-image burden of this one turn. Figure 14 shows that if we give the CoT in an iterative step by step fashion we do not see a major improvement over the single pass vCoT experiments covered in the main paper.

Moreover, we report performance of **MLLMs** using the vCoT in Figure 15. We notice, that **MLLMs** do benefit from visual Chain of Thought significantly more than the tested **UMMs**. We, therefore, believe that trying to equip **UMMs** with the

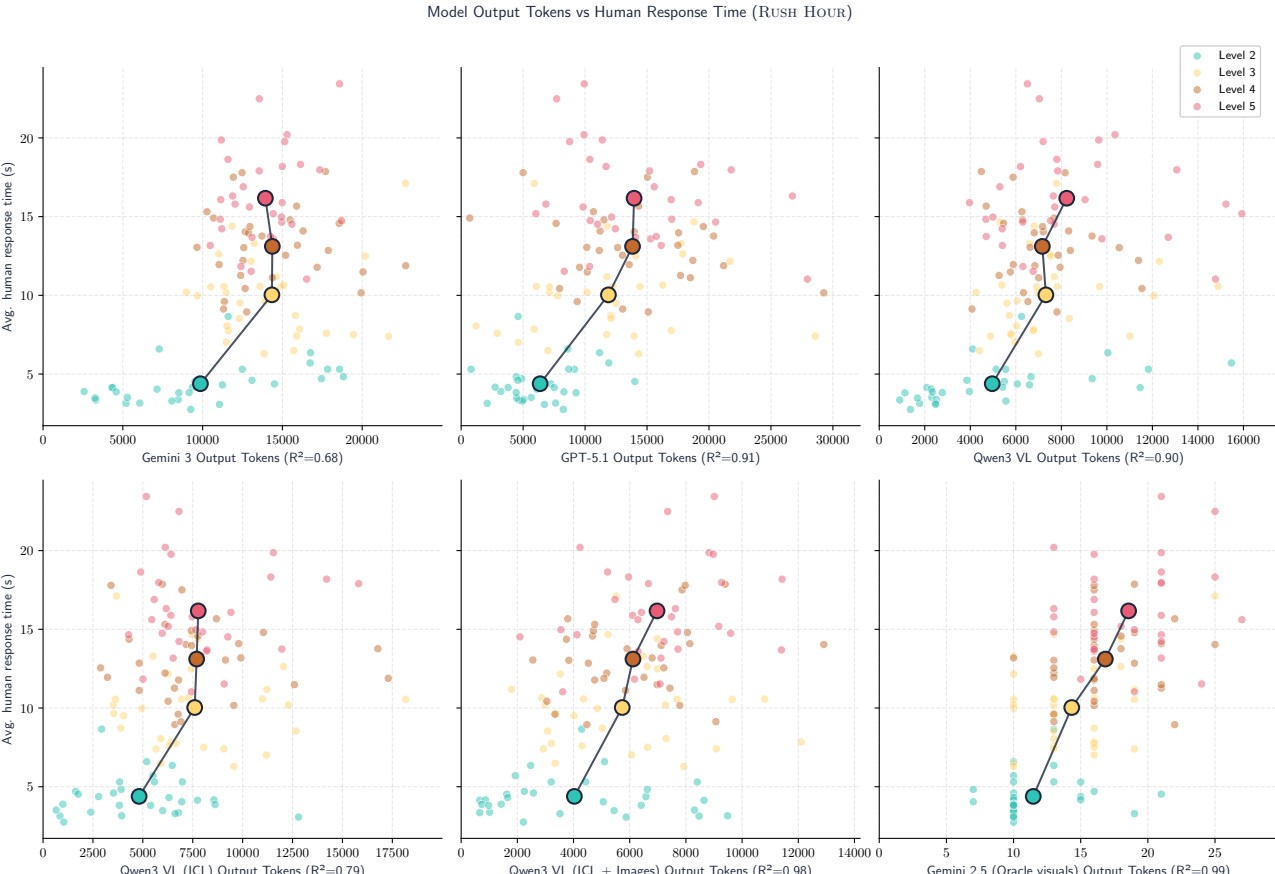

*Figure 11.* **Visual CoT induces linear scaling between model compute and human response time, yet alignment is not a proxy for performance** We find that **Gemini 2.5-I** equipped with oracle visual CoT ($R^2 = 0.99$) and **Qwen3-VL** with in context learning examples containing a visual CoT ($R = 0.98$)—exhibit near-perfect linear scaling, where token expenditure is directly proportional to human cognitive load. However, this alignment is not a silver bullet for accuracy: **Gemini 3** achieves the highest downstream performance on complex tasks (Levels 4–5) despite displaying significantly lower scaling alignment ($R^2 = 0.68$) compared to the other **MLLMs**.

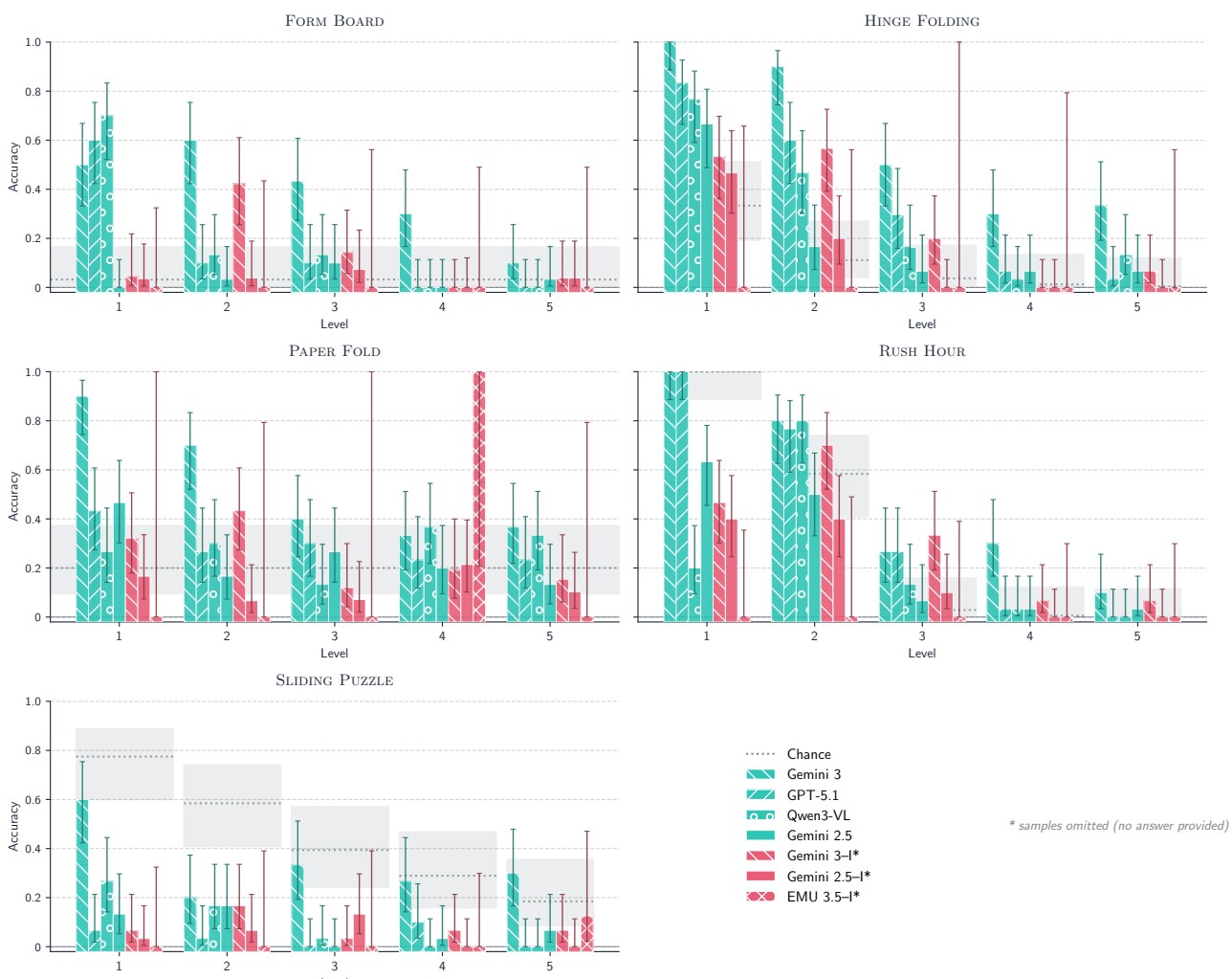

*Figure 12.* **Performance across all difficulty levels for MLLMs and UMMs** While Figure 2 reports results for representative difficulty levels (1, 3, and 5) for clarity, the full set of levels exhibits consistent qualitative trends: performance degrades monotonically with difficulty, and relative differences between model families are stable across adjacent levels.

visual interpretation and reflection ability of MLLMs is a viable next step for further research.

### D.2. Reasoning Budget Comparison on All Tasks

We compare *low* vs. *high* reasoning-budget settings across all MENTISOCULI tasks. Overall, increasing the budget yields negligible and inconsistent changes in accuracy, with differences typically within 95% confidence intervals. Figure 16 reports accuracy by task and difficulty level under both budgets.

## E. Text vs. Latent Decoding

We fine-tune both Mirage and with text-only decoding, i.e. without the latent visual chain-of-thought, on the same 1,000-sample training set(200 samples per level) and with the same Stage 1 hyperparameters as the latent variants (see Appendix A for details). Results are reported in Table 1.

The text-only baseline matches or outperforms the latent variants on every level for LS and on three of five levels for Mirage, despite the latent variants having access to strictly more information (the ground-truth intermediate visuals during training) and a larger compute budget (an additional Stage 2 of 5 epochs for LS and 15 epochs for Mirage). Latent visual decoding therefore does not yield a consistent improvement on RUSH HOUR at the scales we evaluate.

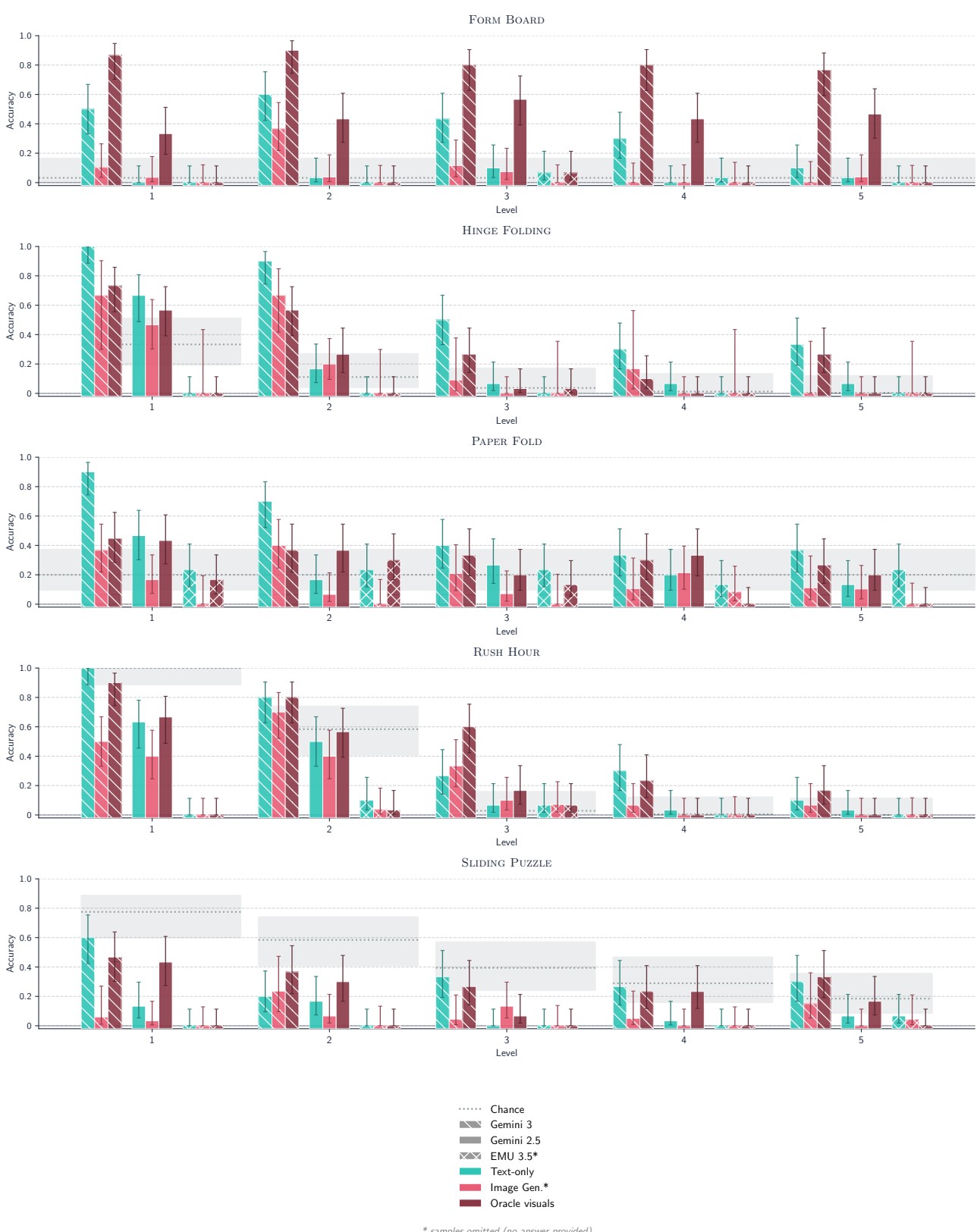

*Figure 13.* **Performance across all difficulty levels of Gemini 2.5, Gemini 2.5-I, Gemini 3, Gemini 3-I and Emu 3.5** While Figure 5 reports results for representative difficulty levels (1, 3, and 5) for clarity, the full set of levels exhibits consistent qualitative trends: benefits from visual intermediates are highly task-dependent and remain consistent across difficulty levels.

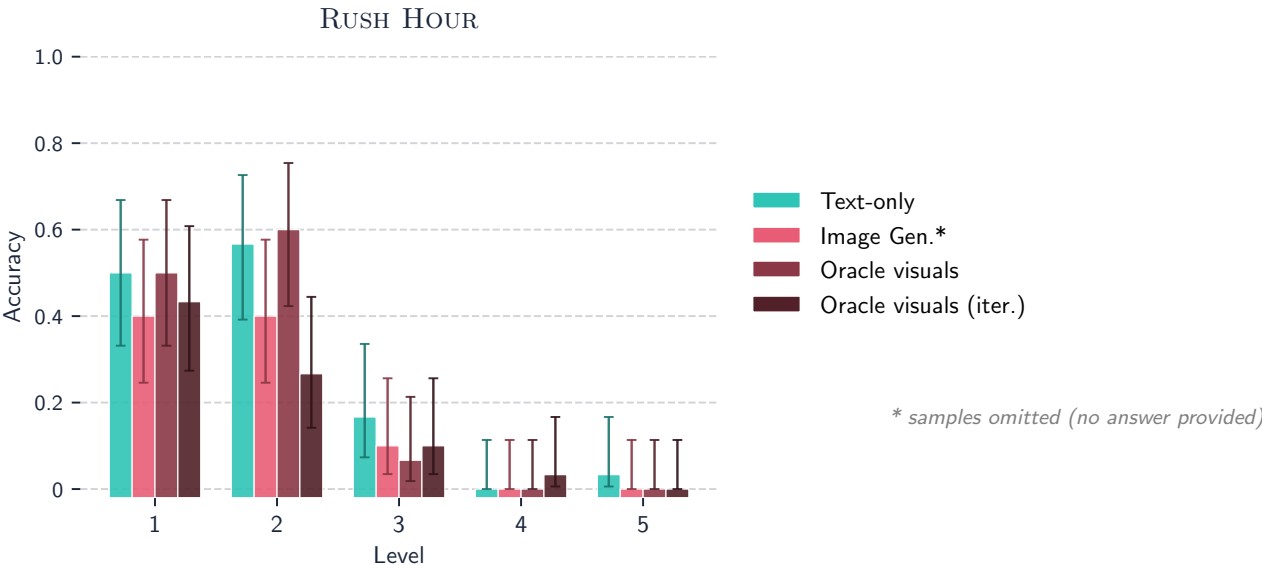

*Figure 14.* **Iterative oracle feedback does not improve performance over standard oracle visuals** We compare standard oracle visuals with an iterative variant that provides intermediate states step-by-step, better matching the information available during generation. Despite this, performance does not improve, indicating that limitations are not due to multi-image input complexity or temporal misalignment.

*Table 1.* **Latent reasoning models do not improve performance over text-only SFT baseline** We finetune the base models of LS and Mirage on 200 samples per level with text-only decoding for a fair comparison. We use the same hyperparameters as during stage 1 for both training setups.

| Level | LS | | Mirage | |
|---|---|---|---|---|
| | w/ latents | text-only | w/ latents | text-only |
| 1 | 93.3% | 100.0% | 100.0% | 100.0% |
| 2 | 66.7% | 76.7% | 96.7% | 93.3% |
| 3 | 6.7% | 23.3% | 50.0% | 43.3% |
| 4 | 0.0% | 10.0% | 26.7% | 40.0% |
| 5 | 0.0% | 0.0% | 10.0% | 16.7% |

# F. Generated Images & Videos

## F.1. Unified Multimodal Models

**Qualitative Results** Qualitative inspection of the visual rollouts from Gemini 2.5-I reveals a pervasive lack of state consistency across all tested domains (Figure 17). Regardless of the task, the model struggles to maintain the identity and geometry of objects between frames. In spatial tasks like PAPER FOLD and HINGE FOLDING, the generated sequences fail to produce a coherent physical history, often introducing nonsensical geometry or extraneous symbols. While rollouts for RUSH HOUR or SLIDING PUZZLE may occasionally show objects moving toward a goal, these sequences are fundamentally undermined by the hallucination of new pieces, the disappearance of others, or illegal changes to the board layout. These errors are not isolated; they compound rapidly, causing the visual state to drift into impossible configurations rather than self-correcting.

Across the two models, Gemini 3-I generally produces visually cleaner and more temporally consistent rollouts than Gemini 2.5-I and Emu 3.5 (Figures 17 to 19). For example, on easier RUSH HOUR levels the generated frames often track the ground-truth state updates closely (see Figure 26), whereas on harder instances the model increasingly hallucinates additional exits, introduces invalid motion directions, or adds/removes cars, with mistakes accumulating over steps. This qualitative trend is consistent with the quantitative improvement from Gemini 2.5-I to Gemini 3-I and Emu 3.5 (Figure 3) and suggests that the performance gain may partially reflect improved state-update image generation (in addition to a stronger base model).

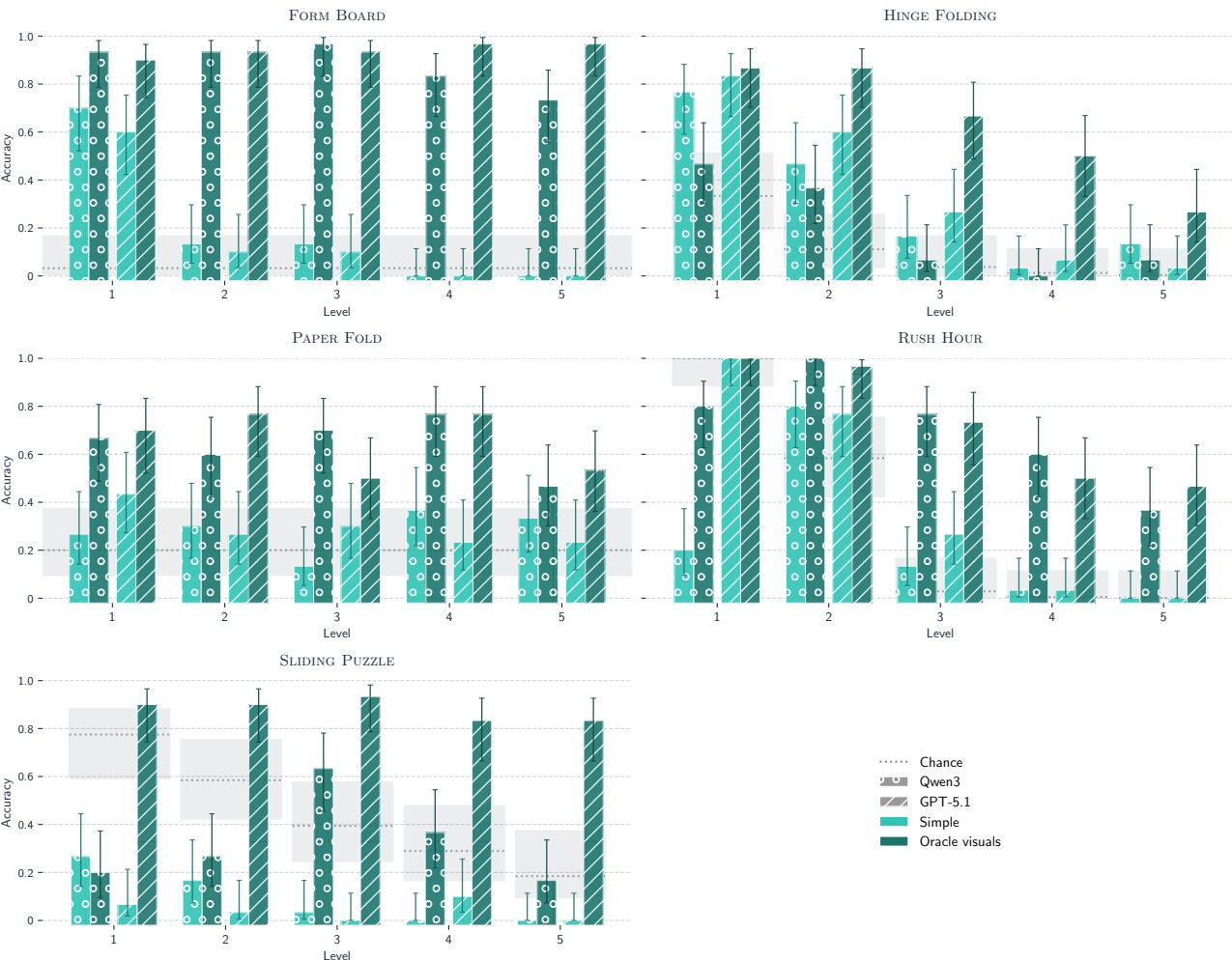

*Figure 15.* **MLLMs can benefit from oracle visual inputs across tasks and difficulty levels** We compare performance with and without oracle visuals that provide correct intermediate states. Both GPT-5.1 and Qwen3 show improvements when given oracle visuals, indicating that strong **MLLM**s can effectively leverage visual information when it is reliable and well-aligned with the task.

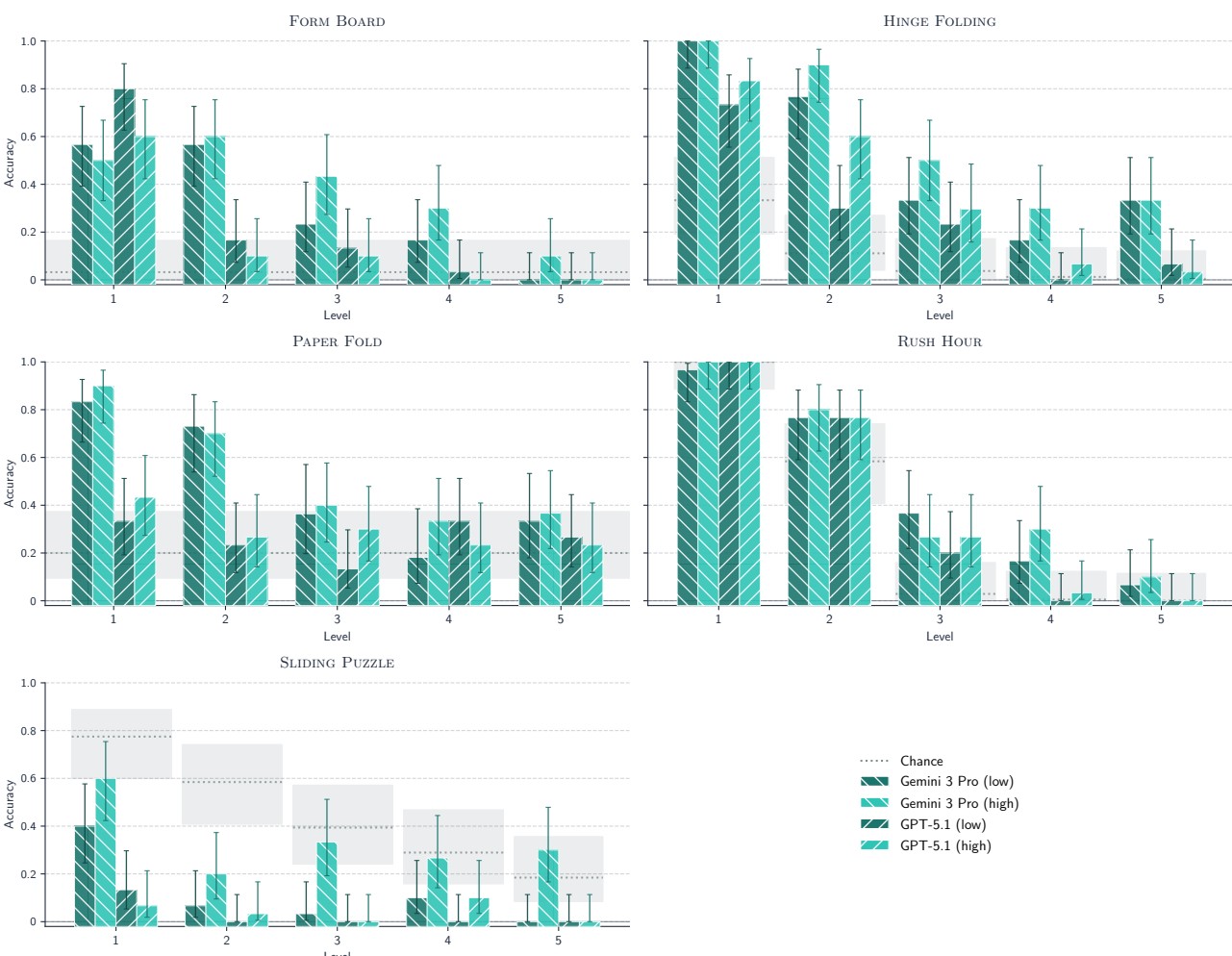

*Figure 16.* **Higher reasoning budget does not reliably improve accuracy** Low vs. high budget results for Gemini 3 and GPT-5.1 across all tasks and levels in MENTISOCULI; any gains are small, inconsistent, and largely disappear at higher difficulty.

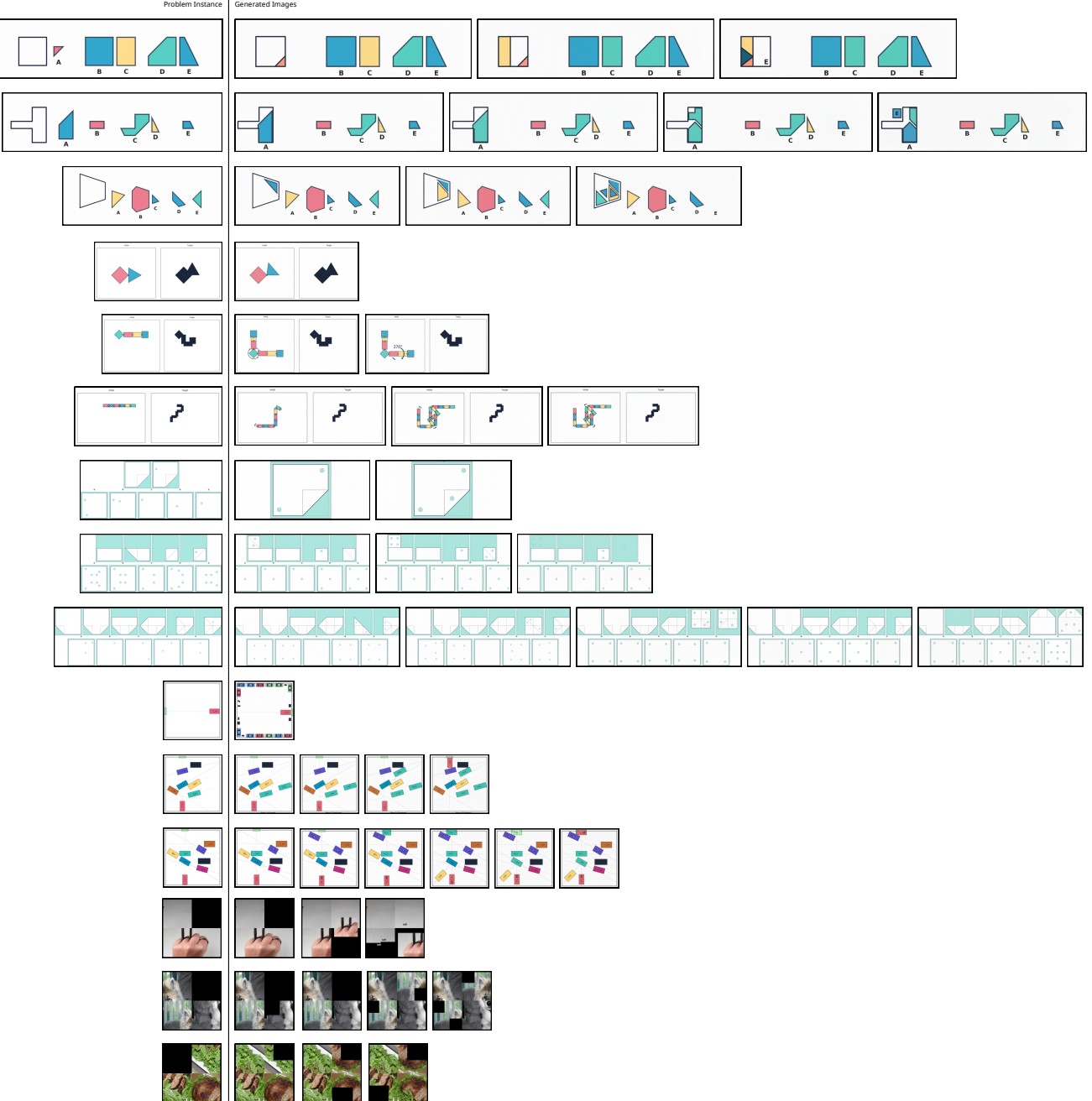

*Figure 17.* Gemini 2.5-I **image rollouts are strongly task-dependent and often drift from valid state updates** Random qualitative samples from instances where the model generated intermediate images (levels 1, 3, and 5), illustrating frequent rule violations and hallucinated state changes in several tasks.

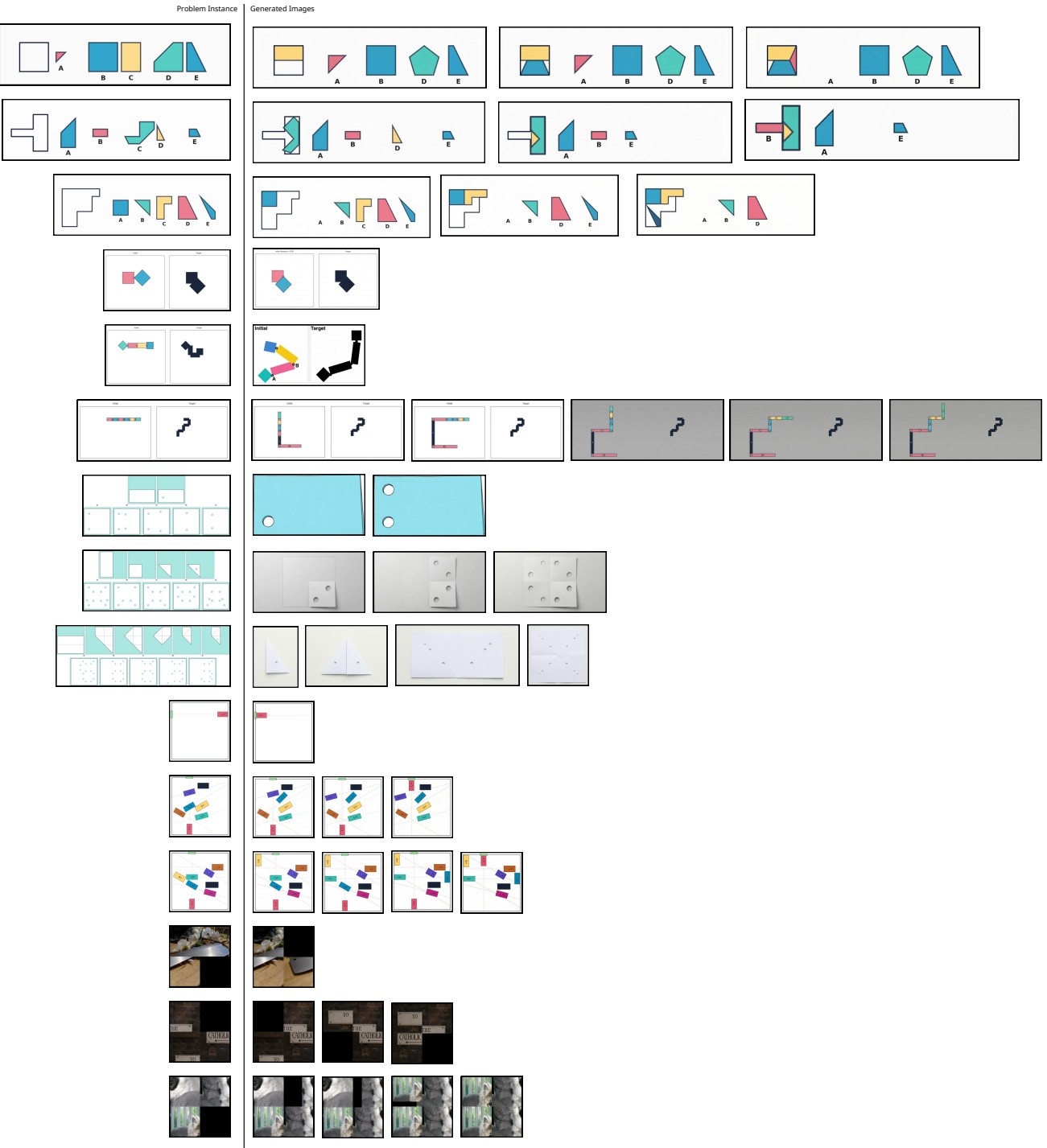

*Figure 18.* **Gemini 3-I produces clean, coherent intermediate states in lower levels, but still hallucinates at higher difficulty** Random qualitative samples from instances with generated intermediate images; highlighting improved visual consistency on easier levels and compounding errors on harder ones.

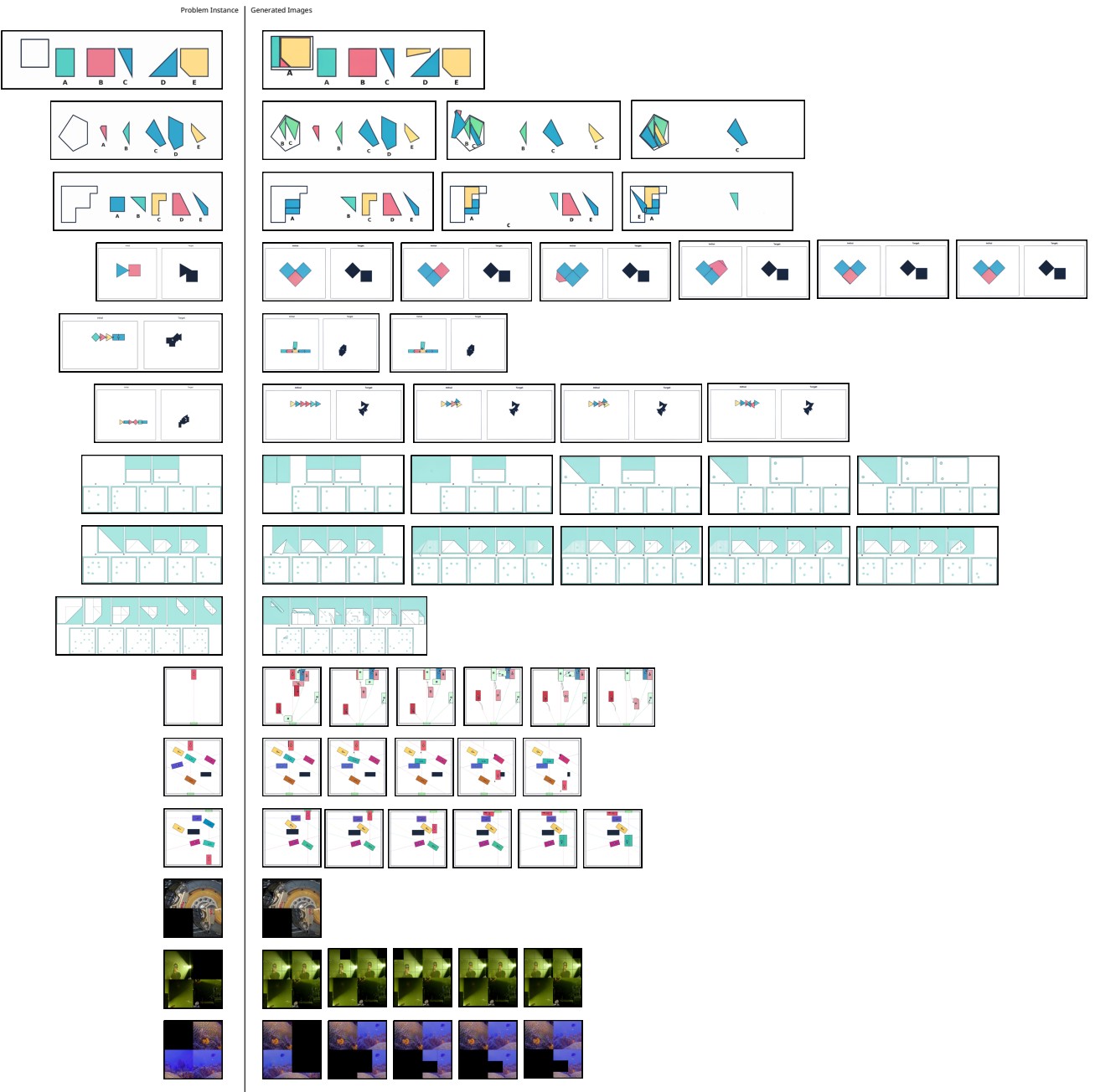

*Figure 19.* **Emu 3.5 produces images that resemble the input image, but do not follow the task rules** Random qualitative samples from instances with generated intermediate images; highlighting that **Emu 3.5** struggles with a basic understanding of task rules.

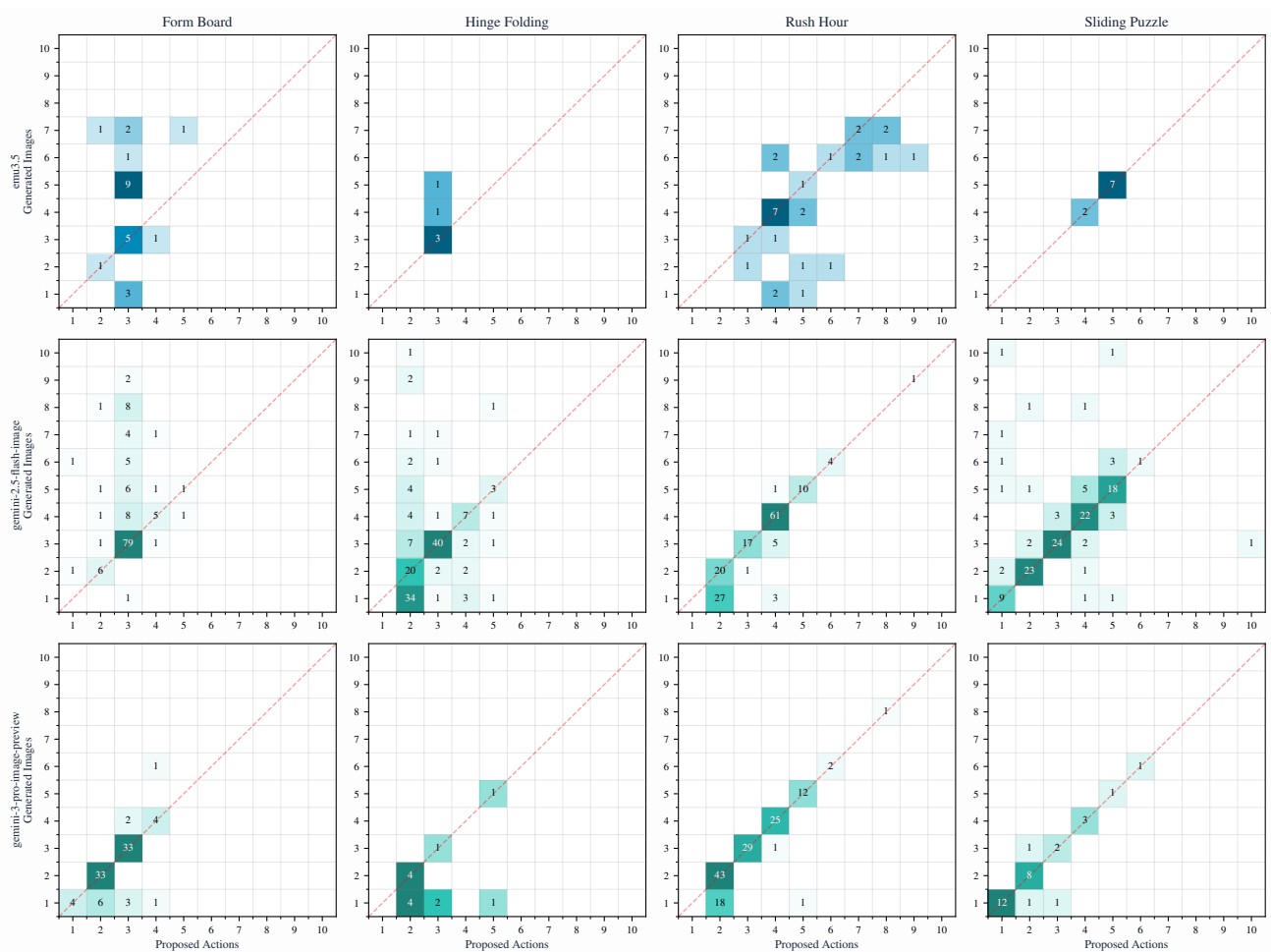

*Figure 20.* **Stronger models show a stronger relationship between generated images and actions** We plot the number of images generated in a rollout against the number of actions proposed by the model; the $x = y$ diagonal represents a strict 1:1 mapping where each move is accompanied by a state update. Samples without generated images are omitted.

**Quantitative Results**   We evaluate the models' visual reasoning via two complementary diagnostics.

First, we assess the faithfulness of state tracking by comparing the number of generated images to the number of proposed actions (Figure 20). An $x = y$ relationship in this context represents an ideal 1:1 mapping where each action is accompanied by an explicit state-update image. We find that **Gemini 3-I** exhibits the highest degree of faithfulness, maintaining a tight diagonal alignment across nearly all tasks. This suggests that when the model commits to a rollout, it consistently produces a systematic image update for every action. In contrast, the alignment for **Gemini 2.5-I** is more task-dependent; while it tracks well in RUSH HOUR, it shows significant vertical variance in FORM BOARD, where the model often produces a disparate number of images for the same number of proposed moves. **Emu 3.5** shows the least coherence in this diagnostic, though its results are less representative due to a lower frequency of image-containing samples.

Second, we measure the models' ability to scale rollout length with instance difficulty by comparing the number of generated images to the *expected* images (derived from ground-truth visual CoT) in Figure 21. **Gemini 3-I** demonstrates the strongest alignment, particularly on lower-difficulty instances. As the expected image count increases, the distribution broadens, suggesting that the model's difficulty estimation or rollout termination becomes less precise at higher complexity. **Gemini 2.5-I** maintains moderate alignment in structured tasks like RUSH HOUR and PAPER FOLD, but its scaling behavior breaks down in SLIDING PUZZLE and FORM BOARD, where difficulty inference is more challenging. Finally, **Emu 3.5** exhibits weak alignment and a general bias toward low generation counts, indicating a failure to adaptively scale its visual reasoning steps to the complexity of the task instance.

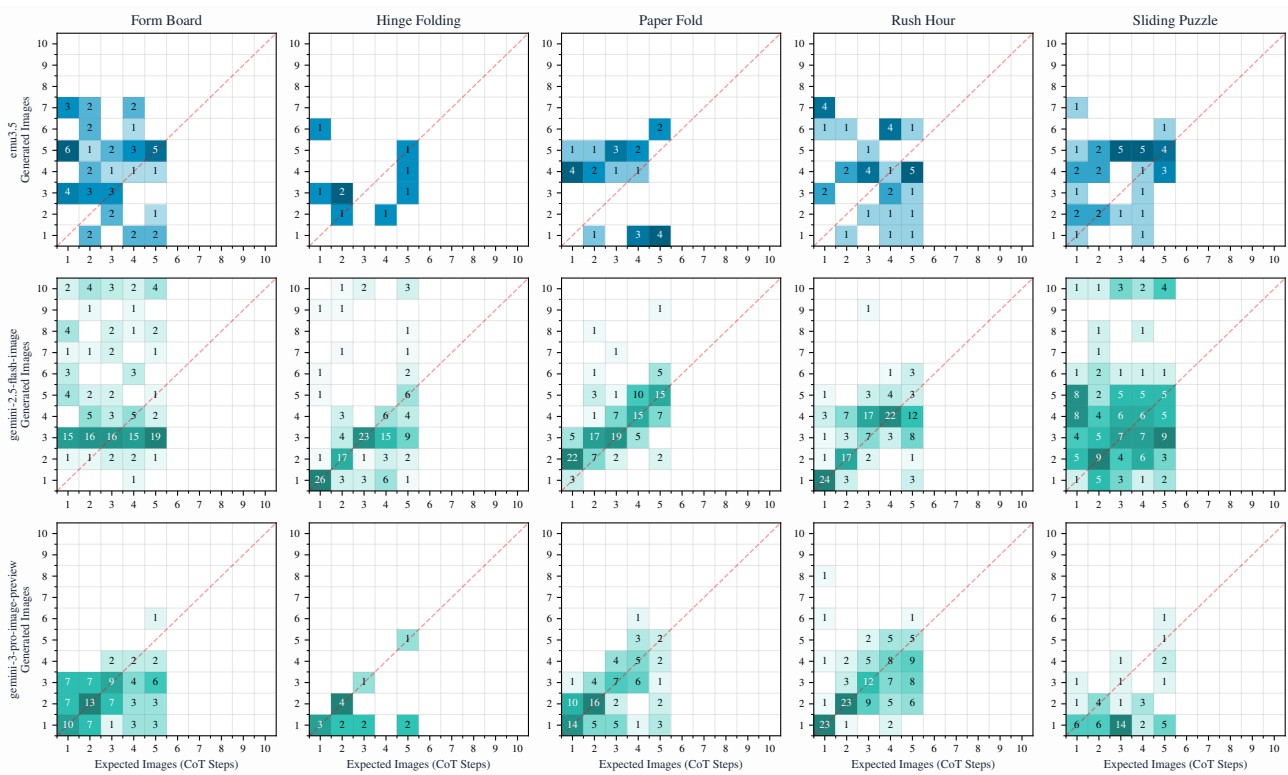

*Figure 21.* **Stronger models show a stronger relationship between generated and expected images** Comparison of the actual number of images produced against the expected image count (the number of intermediate states in the ground-truth visual CoT). This serves as a proxy for whether the model adaptively scales its rollout length to the inherent difficulty of the instance. Samples without generated images are omitted.

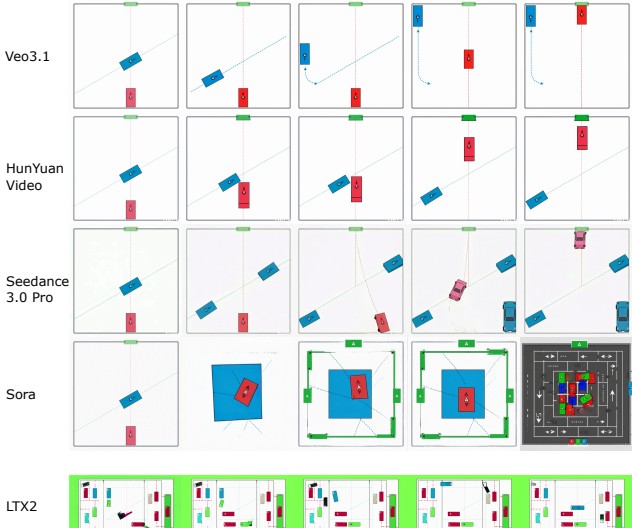

*Figure 22.* **Video frames generated by multiple video models from the same prompt and initial image** Following an initial qualitative comparison, we zoom in on results from Veo-3.1 for more detailed analysis.

### F.2. Qualitative Evaluation of Video Models

We conduct an initial qualitative comparison across multiple open-weight and closed-source video models. The open-weight models include **Hunyuan Video** (Kong et al., 2025), while the closed-source models include **Veo 3.1** (Google DeepMind, 2025a), **Seedance** (Seedance et al., 2025), **LTX-2** (HaCohen et al., 2026), and **Sora** (Liu et al., 2024). Based on the initial comparison as illustrated in Figure 22, we report results primarily for **Veo 3.1**, which consistently produced the most coherent and task-relevant outputs among the tested models.

**Model Selection** We begin by testing each model on RUSH HOUR, which is chosen as a representative tasks. For many models, performance on RUSH HOUR was already substantially suboptimal, and we did not proceed to additional tasks. After initial screening, **Veo 3.1** and **Hunyuan** showed partial attempts at solving RUSH HOUR. We therefore evaluated these models further on all visual reasoning tasks, primarily at Level 2 difficulty. Hunyuan, however, struggled on most tasks beyond RUSH HOUR, whereas Veo 3.1 demonstrated more consistent engagement with the task structure. We include qualitative results from Veo 3.1 on the RUSH HOUR task across five difficulty levels (Figure 23).

**Task-Specific Observations** For SLIDING PUZZLE, RUSH HOUR, and FORM BOARD, **Veo 3.1** appears to readily infer the task setup and produces videos that reflect meaningful attempts at solution execution. However, these attempts remain brittle and cue-driven, and do not translate into consistent task-solving behavior. Performance is sensitive to prompt design: adding stricter constraints and more explicit descriptions of scene structure, such as specifying grid size, improves adherence to task rules. This suggests that the model may rely heavily on textual cues to infer latent spatial structure, rather than robustly integrating visual generation with multi-step reasoning. In contrast, PAPER FOLD proves substantially more challenging. Despite detailed instructions, the model often fails to correctly interpret the spatial relationships in the input image, for example, by treating the entire image as a single sheet of paper or confusing relative spatial references. A plausible explanation is that tasks such as SLIDING PUZZLE, RUSH HOUR, and FORM BOARD are easier because all relevant entities are explicitly visible and can be directly manipulated. PAPER FOLD, by contrast, requires tracking latent spatial structure that is not fully represented in the image, a hypothesis we leave to future work to validate.

## G. Models and Inference details

We query **GPT-5.1** (gpt-5.1-2025-11-13) via the OpenAI API, using sampling parameters temperature $= 1.0$ and top_p $= 1.0$. For experiments that allow tool use, we enable the code_interpreter tool.

We query **Qwen3-VL** (qwen/qwen3-vl-235b-a22b-thinking) via OpenRouter (provider: Nova). The model was

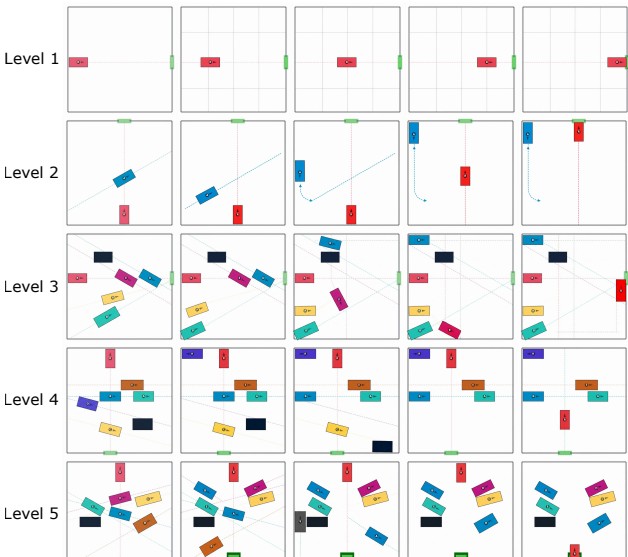

*Figure 23.* **Qualitative results from Veo 3.1 on** RUSH HOUR **across five difficulty levels.**

used in its non-quantized form.

We query **Gemini 2.5** (gemini-2.5-flash), **Gemini 3** (gemini-3-pro-preview), **Gemini 2.5-I** (gemini-2.5-flash-image), **Gemini 3-I** (gemini-3-pro-image-preview), **Veo 3.1** (veo-3.1-generate-preview) both via OpenRouter and directly via the Gemini API. Note that for both **Gemini 2.5** and **Gemini 2.5-I** we query gemini-2.5-flash-image but we only respect answers with or without generated images for the respective model.

For **Mirage** we use the same hyperparameters like the paper, i.e. a latent size of 4, training each stage for 15 epochs, a datasetsize of 1,000 samples distributed uniformly across difficulty levels. For the target image we utilize the last frame of the ground truth visual chain-of-thought. For the text-only ablation we train with the Stage 1 hyperparameters (15 epochs, gradient_accumulation_steps=8) on labels in which the latent visual tokens are removed, leaving a pure text chain-of-thought followed by the final answer.

We fine-tune Qwen2.5-VL-7B-Instruct with the publicly released Latent-Sketchpad Sketch Decoder (sketch_decoder_qwen25_vl.ckpt) on the same 1,000-sample Rush Hour dataset (200 samples per difficulty level). We follow the standard two-stage Latent Sketchpad recipe: **Stage 1** fine-tunes the backbone (LoRA, $r=16$, $\alpha=32$) to predict the sequence of BOI tokens and the answer for 2 epochs at learning rate 1e−4, weight decay 0.01, and effective batch size 16 (per-device batch 1, gradient accumulation 16); **Stage 2** freezes the backbone and trains only the Vision Head for 5 epochs at the same learning rate with effective batch size 16 (per-device batch 8, gradient accumulation 2), using an L1 reconstruction loss against the ground-truth CoT frames. For the text-only ablation we keep the Stage 1 hyperparameters but train on labels that contain only the final \boxed{·} answer (no BOI/EOI tokens, no chain-of-thought text), so the model learns to map the input image directly to the answer with no latent visual reasoning.

### G.1. Omission Rates for UMMs

Table 2 reports the omission rates for all tested **UMMs**. For **Gemini 2.5-I** we query up to three times, for **Gemini 3-I** we inrease to querying up to nine times and for **Emu 3.5** we sample up to 15 times. As seen in Figure 24 this filtering does not bias the answers towards easier / harder traces since including filtered samples in the evaluation does not increase performance.

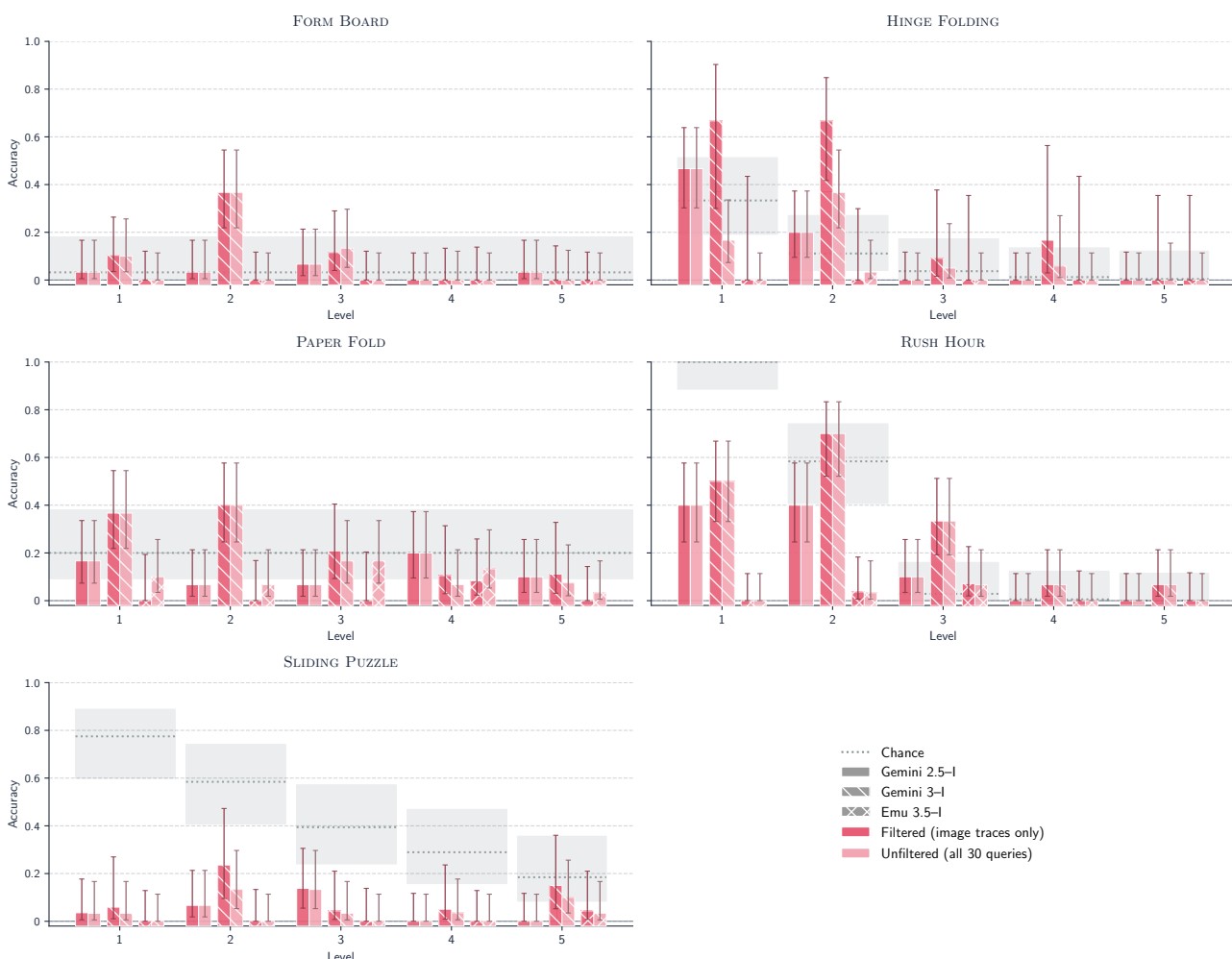

*Figure 24.* **Restricting evaluation to image-based traces does not reduce performance** We compare performance on filtered (image-only traces) and unfiltered (including text-only when no image trace is available) settings across tasks and difficulty levels. Results are nearly identical across conditions, with overlapping error bars and no consistent degradation under filtering, supporting our choice to evaluate exclusively on image-based traces.

| Task | Level | Emu 3.5 | Gemini 2.5-I | Gemini 3-I |
|---|---|---|---|---|
| RUSH HOUR | 1 | 0.0% | 0.0% | 0.0% |
| | 2 | 10.0% | 0.0% | 0.0% |
| | 3 | 6.7% | 0.0% | 0.0% |
| | 4 | 10.0% | 0.0% | 0.0% |
| | 5 | 3.3% | 0.0% | 0.0% |
| SLIDING PUZZLE | 1 | 13.3% | 6.7% | 43.3% |
| | 2 | 16.7% | 0.0% | 43.3% |
| | 3 | 20.0% | 3.3% | 23.3% |
| | 4 | 13.3% | 3.3% | 33.3% |
| | 5 | 23.3% | 3.3% | 33.3% |
| PAPER FOLD | 1 | 46.7% | 0.0% | 0.0% |
| | 2 | 36.7% | 0.0% | 0.0% |
| | 3 | 50.0% | 0.0% | 20.0% |
| | 4 | 20.0% | 0.0% | 36.7% |
| | 5 | 23.3% | 0.0% | 40.0% |
| HINGE FOLDING | 1 | 83.3% | 0.0% | 80.0% |
| | 2 | 70.0% | 0.0% | 50.0% |
| | 3 | 76.7% | 3.3% | 63.3% |
| | 4 | 83.3% | 0.0% | 80.0% |
| | 5 | 76.7% | 3.3% | 76.7% |
| FORM BOARD | 1 | 6.7% | 0.0% | 3.3% |
| | 2 | 3.3% | 0.0% | 0.0% |
| | 3 | 6.7% | 0.0% | 13.3% |
| | 4 | 20.0% | 0.0% | 16.7% |
| | 5 | 3.3% | 0.0% | 23.3% |
| **Overall avg.** | | 28.9% | 0.9% | 27.2% |

*Table 2.* **Most models reliably produce valid interleaved image traces** Reports the fraction of samples where no valid interleaved-image sequence was produced within the retry budget. Gemini 2.5-I almost never fail to produce valid traces across tasks, while Gemini 3-I and Emu 3.5 show higher omission rates, particularly on more complex tasks.

## H. More Examples from MENTISOCULI

### H.1. Example Text Description

The text-only setting uses a simulator-derived *state specification* rather than a human-style natural-language description. Importantly, this representation is *not* isomorphic to a short, low-token text prompt: it is substantially longer, uses continuous-valued attributes (positions, sizes, rotations), and encodes motion constraints explicitly. As a result, solving from text requires models (and humans) to reason over an unusual, high-precision format. See an example of the text description in Figure 25.

### H.2. Example Visual CoT

In addition to the initial rendered instance, MENTISOCULI provides a *ground truth visual chain-of-thought* for each example: a sequence of images meant to aid the visual reasoning process by showing intermediate states. These rollouts serve two purposes in our experiments: they define the *expected* number of intermediate images for an instance and enable direct diagnostics of image-generation-based reasoning by comparing model-generated intermediate states to the ground-truth trajectory (see Figure 26).

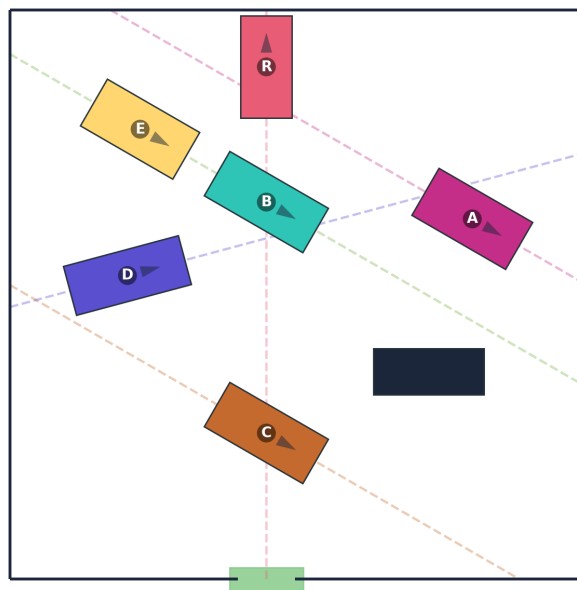

### Text Description

The parking lot has a size of $10 \times 10$.

There is an exit on the bottom ($y = 0$) edge, from $x = 4.01$ to $x = 5.01$.

There is a red car (R) at center $(4.51, 9.00)$ with length $1.80$ and width $0.90$, rotated by $90.0°$, i.e. the car can move forwards along the $(0.00, 1.00)$ axis and backwards along $(-0.00, -1.00)$.
There is a car (A) at center $(8.12, 6.33)$ with length $1.90$ and width $0.95$, rotated by $-30.0°$, i.e. the car can move forwards along the $(0.87, -0.50)$ axis and backwards along $(-0.87, 0.50)$.
There is a car (B) at center $(4.51, 6.62)$ with length $2.00$ and width $0.90$, rotated by $-30.0°$, i.e. the car can move forwards along the $(0.87, -0.50)$ axis and backwards along $(-0.87, 0.50)$.
There is a car (C) at center $(4.51, 2.57)$ with length $2.00$ and width $0.90$, rotated by $-30.0°$, i.e. the car can move forwards along the $(0.87, -0.50)$ axis and backwards along $(-0.87, 0.50)$.
There is a car (D) at center $(2.06, 5.33)$ with length $2.09$ and width $0.89$, rotated by $15.0°$, i.e. the car can move forwards along the $(0.97, 0.26)$ axis and backwards along $(-0.97, -0.26)$.
There is a car (E) at center $(2.29, 7.90)$ with length $1.87$ and width $0.95$, rotated by $-30.0°$, i.e. the car can move forwards along the $(0.87, -0.50)$ axis and backwards along $(-0.87, 0.50)$.

There is a static, immovable object at $((6.38, 3.24), (8.33, 4.05))$.

*Figure 25.* **Text descriptions are verbose simulator states, not compact natural-language prompts** Example RUSH HOUR instance (left) and its deterministic state specification (right), which uses continuous-valued geometry and explicit motion axes.

# I. Prompts & Instructions

## I.1. MLLM Standard Prompts

### FORM BOARD

Look at the image:
It is showing from left to right, a target shape outlined in black and five pieces labeled A through E in various colors:
```
<image combined.png>
```

Rules:
1. The target shape can be assembled using 1 to 5 of the given pieces.
2. Pieces must fit together perfectly with no gaps or overlaps.
3. Some pieces are distractors and are not needed.
4. Pieces are shown in their correct orientation and size (no rotation or scaling needed).

Task:
Determine the subset of pieces from {A, B, C, D, E} necessary to assemble the target shape.

Output: Respond in JSON format as follows:
```
{"answer": "A C E"}
```

List only the letter labels of the pieces needed, separated by space.

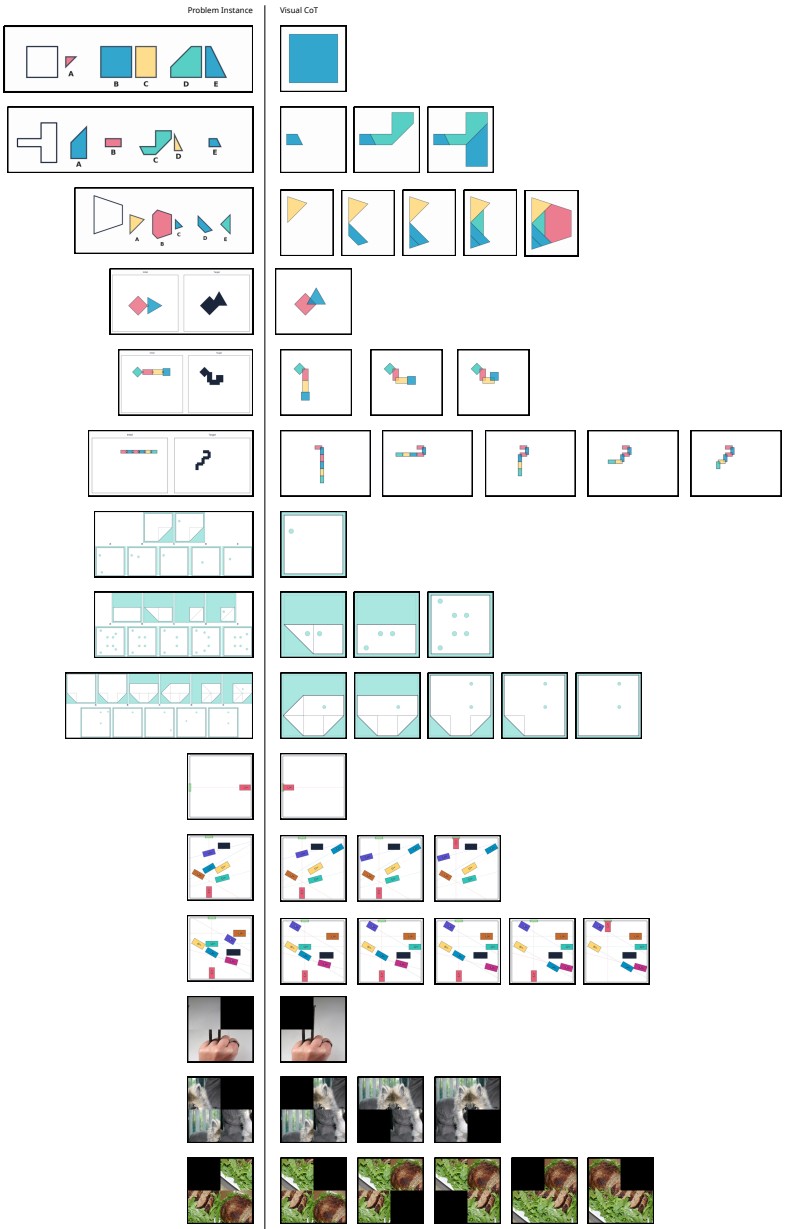

*Figure 26.* **Ground-truth visual CoTs render the simulator state after every action, providing step-aligned supervision for multi-step imagery** Random samples from levels 1, 3, and 5 across tasks; each example shows the initial instance (left) and the corresponding sequence of intermediate rendered states along the reference solution trajectory (right).

## HINGE FOLDING

Look at the image:
The left side shows several rigid shapes connected by labeled hinges (A, B, C, ...).
On the right side is a target folded configuration:
`<image combined.png>`

Rules:
1. Shapes are connected in a kinematic chain; each hinge connects two adjacent shapes.
2. Rotating hinge $N$ causes the shape on the right side of the hinge in the original configuration to rotate anti-clockwise.
   All shapes are connected; all shapes to the right of the rotated shape rotate with it.
   All shapes to the left of the hinge remain fixed.
3. Rotations must be multiples of $90°$ (i.e., $90°, 180°, 270°$).
4. Shapes maintain their connections throughout all rotations.
5. The goal is to find the sequence of hinge rotations that transforms the initial configuration into the target.

Task:
Determine the rotation angle (in degrees) for each numbered hinge to achieve the target configuration.

Output: Respond in JSON format as follows:
`{"answer": "A 90, B 90, C 180"}`

Each pair specifies the hinge label and its rotation angle in degrees.

If multiple solutions exist, output any valid sequence that produces the target configuration.

## PAPER FOLD

Look at the image:
In the first row, it shows a sequence of folds performed on a square paper. The final image in the sequence shows one or more holes being punched into the folded paper.
The second row shows five unfolded square papers with different hole patterns labeled A through E:
`<image combined.png>`

Rules:
1. The paper starts as a flat square.
2. Each step shows the paper being folded along a line (horizontal, vertical, or diagonal).
3. After all folds are complete, one hole is punched through all layers at the marked positions.
4. When the paper is unfolded completely, holes appear at multiple positions due to the layering.
5. One of the five options (A, B, C, D, E) shows the correct hole pattern.

Task:
1. Mentally follow each fold in sequence as shown in the first row.
2. Track where the hole is punched through all folded layers.
3. Mentally unfold the paper with the punched hole step-by-step in reverse order.
4. Determine which unfolded pattern (A, B, C, D, or E) matches your mental result.

Output: Respond in JSON format as follows:

`{"answer": "C"}`

## RUSH HOUR

Look at the image:
It shows the initial configuration of a congested parking lot. Each colored rectangle with a letter and arrow represents a vehicle. Black rectangles without a letter represent immovable obstacles. The light green area at the border indicates the exit. The goal is to move the red vehicle (marked with an R) to the exit:
`<image initial_state.png>`

Rules:
1. Each vehicle can only move forward or backward along its own axis (indicated in the image as a dashed line) — no rotation is allowed. The arrow on each vehicle indicates the forward direction.
2. A vehicle continues to move in the chosen direction until it touches another vehicle, an immovable object, or the image's boundary (marked with a light black line).
3. Only one vehicle moves per action.
4. The red vehicle must reach the exit on the edge of the grid.

Task:
Plan the minimal sequence of moves needed to free the red car and allow it to exit the parking lot.
Each move should specify which vehicle moves and in which direction (forward or backward).

Output: Respond in JSON format as follows:
`{"answer": "A forward, C backward, E forward, R forward"}`

Each pair specifies the vehicle label and the direction of its move.

If multiple sequences lead to a valid solution, output any one valid sequence that allows the red car to exit.

## SLIDING PUZZLE

Look at the image:
Below is a scrambled sliding tile puzzle where a natural image has been cut into an $n \times n$ grid with one blank (black) tile:
`<image initial.png>`

Rules:
1. The puzzle consists of an $n \times n$ grid with one blank tile and $n^2 - 1$ image tiles.
2. You can move the blank tile in four directions: up, down, left, right.
3. Each action swaps the blank with the adjacent tile in the specified direction.
4. Only valid moves are allowed (the blank cannot move outside the grid boundaries).
5. The goal is to reconstruct the original, coherent image by rearranging the scrambled tiles.

Task:
Determine the shortest sequence of moves needed to solve the puzzle and restore the original image.

Output: Respond in JSON format as follows:
`{"answer": "up right down left up"}`

Each word specifies a direction to move the blank tile.

You may guess the most plausible move even if uncertain. Small mistakes are acceptable.

## I.2. Interleaved Image and Text Generation

## FORM BOARD

Look at the image:
It shows, from left to right, a target shape outlined in black and five pieces labeled A through E in various colors:
`<image combined.png>`

Rules:
1. The target shape can be assembled using 1 to 5 of the given pieces.
2. Pieces must fit together perfectly with no gaps or overlaps.
3. Some pieces are distractors and are not needed.
4. Pieces are shown in their correct orientation and size (no rotation or scaling needed).

Task:
Move one piece at a time of {A, B, C, D, E} from the right into the outlined target shape on the left. Generate a new image for each move.
If you notice a mistake, you may also return a piece from the outlined target shape back to the candidate shapes. Also generate a new image in this case.
Finally, using your intermediate images, determine the subset of pieces from {A, B, C, D, E} necessary to assemble the target shape.

Output:
First, reason through the moves and generate images with the updated puzzle states. Make sure that these updated images are generated by the rules specified above. Generate one image after each move.
Finally, respond in JSON format as follows:
`{"answer": "A C E"}`

List only the letter labels of the pieces needed, separated by spaces.

## HINGE FOLDING

Look at the image:
The left side shows several rigid shapes connected by labeled hinges (A, B, C, . . . ).
On the right side is a target folded configuration:
`<image combined.png>`

Rules:
1. Shapes are connected in a kinematic chain — each hinge connects two adjacent shapes.
2. Rotating hinge $N$ causes the shape on the right side of the hinge in the original configuration to rotate anti-clockwise.
   All shapes are connected; all shapes to the right of the rotated shape rotate with it.
   All shapes to the left of the hinge remain fixed.
3. Rotations must be multiples of $90°$ (i.e. $90°$, $180°$, $270°$).
4. Shapes maintain their connections throughout all rotations.
5. The goal is to find the sequence of hinge rotations that transforms the initial configuration into the target.

Task:
Determine the rotation angle (in degrees) for each numbered hinge to achieve the target configuration.
After each move, generate a new image in which you update the left side of the image to reflect your proposed hinge rotation.
Once the outlines of the left and right side of the image match, output your rotation sequence.

Output:
First, reason through the moves and generate images with the updated puzzle states. Make sure that these updated images are generated by following the rules specified above. Generate one image after each rotated hinge.
Respond in JSON format as follows:
`{"answer": "A 90, B 90, C 180"}`

Each pair specifies the hinge label and its rotation angle in degrees.

If multiple solutions exist, output any valid sequence that produces the target configuration.

## PAPER FOLD

Look at the image:
In the first row, it shows a sequence of folds performed on a square paper. The final image in the sequence shows one or more holes being punched into the folded paper.
The second row shows five unfolded square papers with different hole patterns labeled A through E:
`<image combined.png>`

Rules:
1. The paper starts as a flat square.
2. Each step shows the paper being folded along a line (horizontal, vertical, or diagonal).
3. After all folds are complete, one hole is punched through all layers at the marked positions.
4. When the paper is unfolded completely, holes appear at multiple positions due to the layering.
5. One of the five options (A, B, C, D, E) shows the correct hole pattern.

Task:
1. Mentally follow each fold in sequence as shown in the first row.
2. Track where holes are punched through all folded layers.
3. Unfold the paper with the punched hole step-by-step in reverse order.
4. After each unfolding move, generate an image of the (partially) unfolded paper with hole(s) in the correct positions.
5. Determine which unfolded pattern (A, B, C, D, or E) matches your generated result.

Output:

First, reason through the moves and generate images with the progressively more unfolded paper sheet. Make sure that these updated images are generated by following the rules specified above.

Generate one image after each unfold.

Finally, respond in JSON format as follows:

`{"answer": "C"}`

## RUSH HOUR

Look at the image:
It shows the initial configuration of a congested parking lot. Each colored rectangle with a letter and arrow represents a vehicle. Black rectangles without a letter represent immovable obstacles. The light green area at the border indicates the exit. The goal is to move the red vehicle (marked with an R) to the exit:
`<image initial_state.png>`

Rules:
1. Each vehicle can only move forward or backward along its own axis (indicated in the image as a dashed line) — no rotation is allowed. The arrow on each vehicle indicates the forward direction.
2. A vehicle continues to move in the chosen direction until it touches another vehicle, an immovable object, or the image's boundary (marked with a light black line).
3. Only one vehicle moves per action.
4. The red vehicle must reach the exit on the edge of the grid.

Task:
Plan the minimal sequence of moves needed to free the red car and allow it to exit the parking lot.
Each move should specify which vehicle moves and in which direction (forward or backward).
After each move, generate an image showing the updated puzzle state.

Output:
First, reason through the moves and generate images with the updated puzzle states. Make sure that these updated images are generated by following the rules specified above.
Generate one image after each move.
Finally, respond in JSON format as follows:
`{"answer": "A forward, C backward, E forward, R forward"}`

Each pair specifies the vehicle label and the direction of its move.

If multiple sequences lead to a valid solution, output any one valid sequence that allows the red car to exit.

## SLIDING PUZZLE

Look at the image:
Below is a scrambled sliding tile puzzle where a natural image has been cut into an $n \times n$ grid with one blank (black) tile:
`<image initial.png>`

Rules:
1. The puzzle consists of an $n \times n$ grid with one blank tile and $n^2 - 1$ image tiles.
2. You can move the blank tile in four directions: up, down, left, right.
3. Each action swaps the blank with the adjacent tile in the specified direction.
4. Only valid moves are allowed (the blank cannot move outside the grid boundaries).
5. The goal is to reconstruct the original, coherent image by rearranging the scrambled tiles.

Task:
Determine the shortest sequence of moves needed to solve the puzzle and restore the original image.
After each move, generate an image of how the puzzle state looks, i.e., showing the swap of the blank tile and the adjacent image tile.

Output:
First, reason through the moves and generate images with the updated puzzle states. Make sure that these updated images are generated by following the rules specified above. Generate one image after each proposed move.
Finally, respond in JSON format as follows:
`{"answer": "up right down left up"}`

Each word specifies a direction to move the blank tile.

You may guess the most plausible move even if uncertain. Small mistakes are acceptable.

## I.3. Video Models

---
**RUSH HOUR**

Look at the image:
Below is the first image, showing the initial configuration of a congested parking lot. Each colored rectangle represents a vehicle, and the red car is the one that must reach the exit. The exit is marked with a green area at the border:

Rules:
1. Each vehicle can only move forward or backward with straight sliding motion along its own axis.
2. No rotation is allowed at any time.
3. A vehicle continues to move in the chosen direction until it touches another vehicle or a boundary.
4. Only one vehicle moves per action.
5. The goal is for the red car to reach the exit located on the edge of the grid.
6. Vehicle shapes, colors, exit, and outlines must not change throughout the solution.
7. No camera motion: no zoom, no pan, no rotate, no tilt, no dolly.
8. Do not add or remove anything: no new objects, labels, lights, shadows, reflections, textures, markings, or UI elements.
9. The background, grid, exit, and all pieces remain perfectly static, except for the piece currently sliding.

Task:
Plan the minimal sequence of moves needed to free the red car and allow it to exit the parking lot.

Output:
A video demonstrating the full solution to the puzzle, one move at a time.

```
<image initial_state.png>
```
---

## I.4. Human Instructions

For the human psychophysics experiment we used the following set of instructions:

---
**Human Instructions**

Look at the image:
You are shown the initial configuration of a congested parking lot.
Each colored rectangle with a letter and arrow represents a vehicle. Black rectangles without a letter represent immovable obstacles. The light green area at the border indicates the exit. The goal is to move the red vehicle (marked with an R) to the exit.
```
<image initial_state.png>
```

Rules:
1. Each vehicle can only move forward or backward along its own axis (indicated in the image as a dashed line) — no rotation is allowed. The arrow on each vehicle indicates the forward direction.
2. A vehicle continues to move in the chosen direction until it touches another vehicle, an immovable object, or the image's boundary (marked with a light black line).
3. Only one vehicle moves per action.
4. The red vehicle must reach the exit on the edge of the grid.

Task:
Plan the minimal sequence of moves needed to free the red vehicle and allow it to exit the parking lot.
Each move should specify which vehicle moves and in which direction (forward or backward).

Output:
Respond by first specifying the label of the vehicle, then specifying whether it should move forwards or backwards, for example:
```
AF CB RF
```
to indicate that first A should move forward, then C should move backward, and then the red vehicle should move forward, i.e. each pair specifies the vehicle label and the direction of its move.
Once you are done, press ENTER.
If multiple sequences lead to a valid solution, output any one valid sequence that allows the red car to exit.

Press ENTER to begin the experiment.

---

## I.5. Ground Truth Visual Chain of Thought

We append the prompt from Appendix I.1 with

---
**GT Visual CoT**

The following images correspond to intermediate images in the reasoning process.

You must use them to obtain your answer:

```
<images visual cot>
```
---

## I.6. Tool Use

We append the prompt from Appendix I.1 with

> **Tool Use**
>
> Use your python tool to solve this question.

## I.7. In-Context Learning

We append the prompt from Appendix I.1 with one example from each level. We use the same examples for all queried models across in-context learning (ICL) with intermediate images and no intermediate images.

> **In-Context Learning Prompt (no intermediate visuals). Shortened to fit on one page**
>
> Examples:
>
> 1. This is the initial parking lot:
>
> 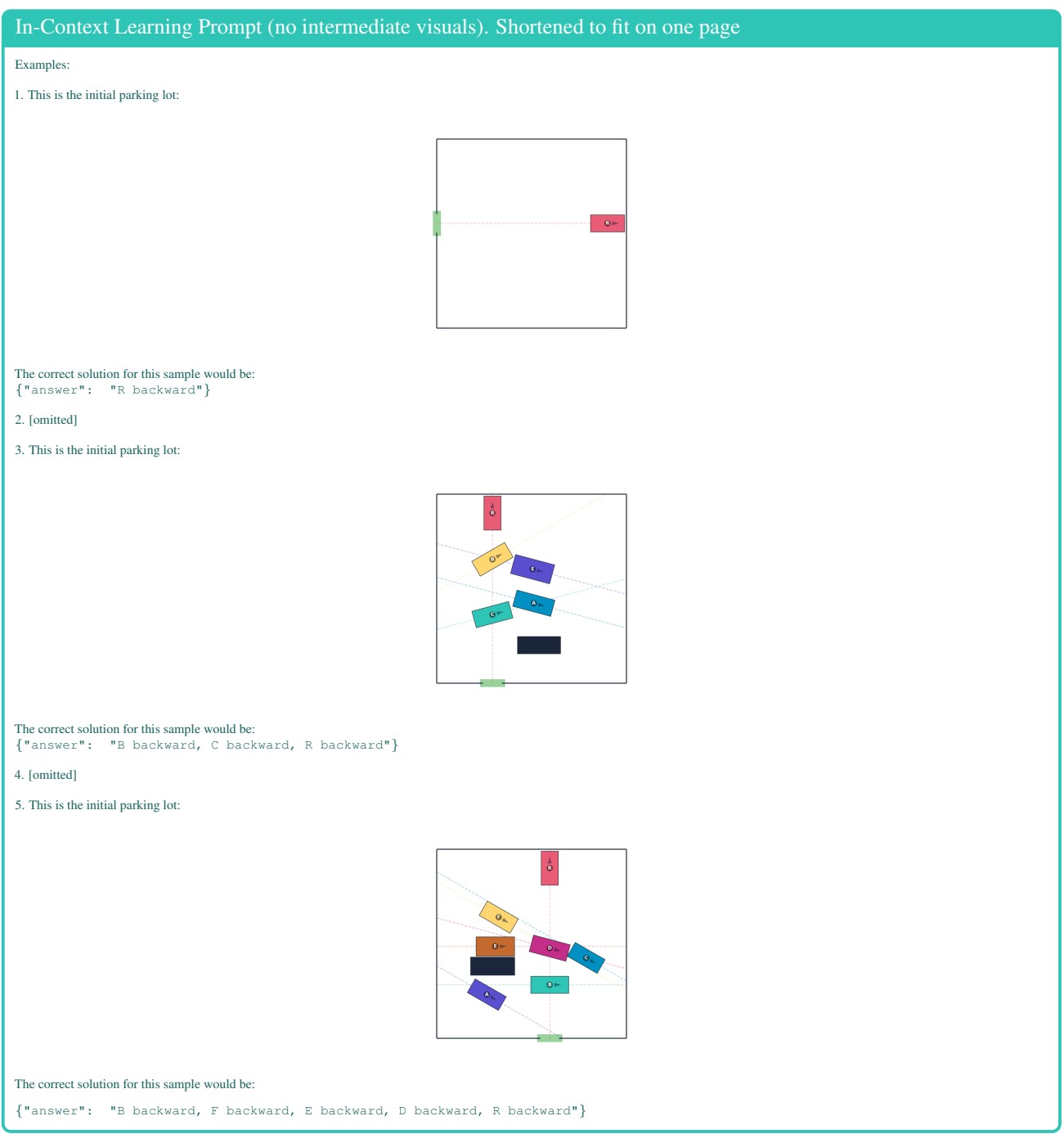
>
> The correct solution for this sample would be:
> ```
> {"answer":  "R backward"}
> ```
>
> 2. [omitted]
>
> 3. This is the initial parking lot:
>
> The correct solution for this sample would be:
> ```
> {"answer":  "B backward, C backward, R backward"}
> ```
>
> 4. [omitted]
>
> 5. This is the initial parking lot:
>
> The correct solution for this sample would be:
>
> ```
> {"answer":  "B backward, F backward, E backward, D backward, R backward"}
> ```

In-Context Learning Prompt (intermediate visuals). Shorten to fit on a page

Examples:

1. [omitted]

2. [omitted]

3. This is the initial parking lot:

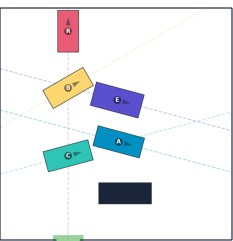

After moving B backward the parking lot would look like:

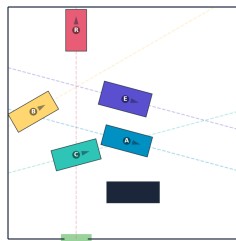

After moving C backward the parking lot would look like:

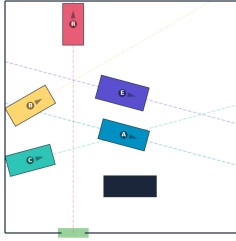

After moving R backward the parking lot would look like:

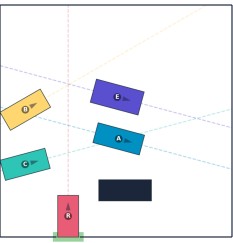

The correct solution for this sample would be:
```
{"answer":  "B backward, C backward, R backward"}
```

4. [omitted]

5. [omitted]

## I.8. Optimized Prompt



### Optimized Prompt

Look at the image:
You are given an image of a RUSH HOUR–style sliding block puzzle. Each colored rectangle with a capital letter and an arrow is a movable vehicle. Black rectangles with no letters are fixed obstacles (walls). A light green opening on the border is the exit. The goal is to move the red vehicle labeled "R" so it can slide out through the exit:
`<image initial_state.png>`

Rules:
1. Each vehicle is either horizontal or vertical (shown by its arrow direction). Vehicles never rotate or move diagonally.
2. A vehicle may move only along its own axis:
   – "forward" = exactly in the direction the arrow points,
   – "backward" = exactly opposite that direction.
3. When a vehicle moves, it must slide in the chosen direction until its front edge first touches:
   – another vehicle, or
   – a fixed black obstacle, or
   – the outer boundary of the grid.
   It cannot stop earlier and cannot pass through anything.
4. Only one vehicle moves per action.
5. The puzzle is solved when R can make a single legal slide through the green exit area.

Task:
1. From the image, internally identify every vehicle by:
   – its letter (a single uppercase letter),
   – its arrow direction (which defines forward vs. backward),
   – its orientation (horizontal or vertical).
2. Internally reason step-by-step to find a legal sequence of moves that:
   – clears a path for R to the exit, and then
   – moves R out through the exit in a final legal move.
   Focus first on vehicles directly blocking R, then on vehicles blocking those blockers, and so on. Avoid pointless back-and-forth moves; prefer short, efficient solutions.
3. Check that every planned move obeys the rules: the chosen vehicle moves only along its axis and slides as far as possible in that direction until contact.

Output format:
– Do all visual analysis and reasoning internally. Do NOT display your intermediate reasoning or any text other than the JSON object.
– Respond with a single JSON object and no extra text before or after it.
– Use exactly this format: one string listing the moves in order, separated by commas:
`{"answer": "A forward, C backward, E forward, R forward"}`

Each item in the string must be:
– a single uppercase vehicle letter from the image,
– a space,
– the word "forward" or "backward".
Return only this JSON object as your final answer.



