# OpenReview forum: "MentisOculi: Revealing the Limits of Reasoning with Mental Imagery"
_ICML.cc/2026/Conference — ICML 2026 regular_

### Official Review · Reviewer_xNHQ · 2026-02-26

**Soundness:** 3
**Presentation:** 3
**Significance:** 3
**Originality:** 3
**Overall Recommendation:** 4
**Confidence:** 5

**Summary:**

The paper introduces MENTISOCULI, a procedurally generated, stratified benchmark designed to evaluate the capacity of multimodal models to utilize "mental imagery" for multi-step visual reasoning. The authors evaluate a broad spectrum of models, revealing that explicit, self-generated visual thoughts currently fail to yield consistent performance gains over text-only reasoning.

**Compliance With Llm Reviewing Policy:**

Affirmed.

**Final Justification:**

I thank the authors for their careful and constructive rebuttal. They addressed my main questions during the review process and clarified the aspects I had previously found uncertain. In my view, the work remains a solid contribution with clear motivation, reasonable technical soundness, and meaningful empirical value. Since my initial assessment was already positive, the rebuttal mainly reinforced rather than changed my evaluation. I therefore maintain my recommendation of Weak Accept.

**Key Questions For Authors:**

1. The paper provides a compelling diagnosis of "generation errors" and "interpretation errors" in current UMMs. Looking forward, what are your insights on the most promising paths to bridge this gap?
2. Your evaluation highlights that explicit, self-generated imagery currently fails to consistently improve reasoning. Given these limitations, how do you view the future trajectory of explicit pixel-space generation compared to latent visual reasoning paradigms (e.g., Mirage, Latent Sketchpad)? Which paradigm do you anticipate holds more potential for scalable, compounding multi-step reasoning?

**Limitations:**

yes

**Strengths And Weaknesses:**

## Strengths:
1. Insightful Error Disentanglement: The paper makes a valuable contribution to the unified modeling of multi-modal generation and comprehension by formally decoupling "generation errors" from "interpretation errors". By prompting models with oracle visual chains-of-thought, the authors demonstrate that UMMs fail not only at generating coherent visual states, but also at interpreting and grounding their decisions in ground-truth visual evidence.
2. Comprehensive Paradigm Comparison: The evaluation spans the entire spectrum of machine mental imagery. Testing text-only MLLMs against models that interleave explicitly generated images, models utilizing latent visual tokens, and natively visual video models provides a comprehensive assessment.
3. Rigorous and Purpose-Built Task Design: The benchmark is meticulously crafted to isolate true reasoning with "mental imagery" rather than evaluating basic visual comprehension. By enforcing high information density and dynamic geometric constraints, the tasks effectively neutralize trivial text-based shortcuts.

## Weaknesses:
1. Narrow Scope of Latent Reasoning Baselines: The evaluation of latent visual thought elicitation is somewhat limited. The paper relies exclusively on Mirage, which may underrepresent the potential of sketching visual thoughts in a latent space. Furthermore, although relatively recent, other highly relevant latent reasoning frameworks, such as Latent Sketchpad[1], should be discussed to provide a more comprehensive view.
2. Lack of Proposed Mitigation: As an evaluation paper, this work does an well job of diagnosing the problem, so the lack of a proposed mitigation is not a major flaw. However, the paper would benefit from a richer discussion on how the community might overcome these hurdles.

[1] Zhang H, Wu W, Li C, et al. Latent sketchpad: Sketching visual thoughts to elicit multimodal reasoning in mllms[J]. arXiv preprint arXiv:2510.24514, 2025.

---

> ### Author Rebuttal · Authors · 2026-03-31
>
> We thank the reviewer for the highly encouraging feedback and for recognizing the value of our error disentanglement and the rigorous design of the MENTISOCULI benchmark. We appreciate the opportunity to expand on the future trajectories of visual reasoning.
> We address your questions below:
>
> **[W1 – Narrow Scope of Latent Reasoning Baselines]** Thank you for pointing us to Latent Sketchpad (LS). We evaluate LS on Rush Hour using Qwen 2.5-VL fine tuning on 200 samples per level (like in Mirage), prepending the images (this time one per step) and then making the model output the answer. Preliminary results are in the Table https://ibb.co/gZgTyyqD and a qualitative example in Figure https://ibb.co/G3nMZ7pQ .
> After rebuttal we plan to add performance of text-only decoding of the same model for a fair comparison across model sizes.
>
> **[W2, Q1, Q2 – Expanded Discussion Section]**
> We thank the reviewer for encouraging a deeper discussion of future directions and are happy to see that our work sparked an interest in further discussions.
> Looking forward, we believe that any paradigm aiming to reliably solve model mental imagery must overcome these core bottlenecks:
>
> Models must produce consistent and valid visual rollouts to support planning. In our experiments, all paradigms—including UMMs, video models, and (the decoded output of) latent reasoning approaches—suffered from generation errors, leading to drifting or inconsistent states. We expect any mitigation that addresses hallucinations and generation errors, or improved prompt-following in image/video generation to improve the utility of self-generated imagery for reasoning. These goals align well with general media generation desiderata.
>
> Beyond generating visual states, models must use them as actionable, causal evidence. A strong link must exist between the generated image and the textual reasoning. For this link to be meaningful, a model must be able to interpret its imagined states and, crucially, backtrack or adjust if a rollout reveals a flawed plan. UMMs currently struggle with interpretation errors, frequently failing to ground their decisions in the images they just explicitly generated (as detailed in our response to Reviewer k7o9). Video models natively generate rich rollouts but currently lack the autoregressive planning mechanisms necessary to adjust course based on intermediate frames. We expect this problem might be partially addressed by including more relevant multi-turn image-text data in all training stages of future UMMs. Data for multimodal problem solving is likely underrepresented in current training pipelines, as model developers seem to focus more on general generation quality. Our results invite the community consider other use-cases of native media generation.
>
> Finally, latent approaches can learn representations that partially mitigate interpretation errors. While our current experiments with latent models do not yet show explicit self-correction or backtracking based on imagined states, we believe this is likely a consequence of smaller model sizes and strong task-specific fine-tuning. The development of larger, more capable latent models is an exciting prospect for machine mental imagery.
>
> Ultimately, our findings suggest the core challenge is not just generating or perceiving visuals, but learning representations that enable stable state transitions for reliable multi-step planning. We hypothesize that the most promising path forward is a hybrid approach: coupling a strong visual world model (for accurate state rollouts) with a robust planning/search module (to evaluate imagined actions and backtrack when necessary).
>
> We will expand the Discussion section in the revised camera-ready manuscript to explicitly incorporate these perspectives and future trajectories.

---

> > ### Author Rebuttal · Reviewer_xNHQ · 2026-04-01
> >
> > Thank you for your detailed response. Given that my initial assessment was already positive, I will maintain my current score.

---

### Official Review · Reviewer_d3cr · 2026-03-12

**Soundness:** 2
**Presentation:** 3
**Significance:** 2
**Originality:** 3
**Overall Recommendation:** 4
**Confidence:** 4

**Summary:**

This paper investigates whether advanced artificial intelligence models can improve their reasoning ability by leveraging intermediate images, a process similar to human mental visualization. To test this capability, the authors created a new benchmark called MentisOculi, which consists of complex, multi-step visual puzzles. Their experiments reveal that current multimodal models fail to leverage intermediate visual generation to improve reasoning performance.

**Compliance With Llm Reviewing Policy:**

Affirmed.

**Final Justification:**

The newly included experiments and analysis address my concerns. So I increase the score to 4.

**Key Questions For Authors:**

1. Could you please share the text output in figure 15 to demonstrate the complete reasoning process?

**Limitations:**

yes

**Strengths And Weaknesses:**

**Strengths:**

1. The paper is well-written, clearly structured, and easy to follow.
2. The exploration of mental imagery and visual Chain-of-Thought (CoT) is a timely and compelling research topic.
3. The inclusion of a human baseline and chance performance metrics establishes an effective foundation for the analysis. Furthermore, the findings regarding adaptive reasoning effort are highly insightful.

**Weaknesses:**

1. **Unconvincing "UMM" Evaluation:** Categorizing these specialized models as UMMs lacks rigor. Both Gemini-2.5-I (Nano Banana) and Gemini-3-I (Nano Banana Pro) are officially positioned as image generation and editing models. Because these models are explicitly optimized for image generation, **evaluating their visual outputs would be a more natural and appropriate metric than evaluating their text outputs.**
2. **Comparison of UMM and MLLM Counterparts:** Lines 247-248 compare the performance of UMMs against their MLLM counterparts. However, there is insufficient public information regarding the architectural and implementation differences between Gemini-3 and Gemini-3-I (Nano Banana Pro), as well as between Gemini-2.5 and Gemini-2.5-I (Nano Banana), making this a difficult comparison to validate. Furthermore, as stated in lines 1171-1175, the experimental setup is inherently inconsistent: the Gemini 2.5 and Gemini 2.5-I evaluations both query the exact same underlying model (`gemini-2.5-flash-image`), whereas the Gemini 3 and Gemini 3-I evaluations utilize distinct, independent models. This inconsistent configuration confounds the results and is highly confusing.
3. **Ablation on Ground-Truth Visual CoT:** Appending the entire ground-truth visual CoT to the input in a single pass effectively reduces the experiment to a standard multi-image understanding task, which deviates from the core problem the authors are attempting to solve. Instead, a progressive approach should be considered. For example, the methodology would be stronger if it provided one CoT image per user turn, prompting and evaluating the MLLM's response at each intermediate step.

---

> ### Author Rebuttal · Authors · 2026-03-31
>
> We thank you for the constructive feedback and for the "highly insightful" designation of our adaptive reasoning experiments. We address your feedback regarding model categorization and the consistency of our evaluation below.
>
> **[W1 – Evaluating UMMs on Generated Images]** We evaluate the generated image sequences from all tested UMMs (Gemini 3 I, Gemini 2.5 I and Emu 3.5) in Figure https://ibb.co/B5YrbVmm . For this we adjusted our video auto-rater to work with image sequences. Per adjacent images we expect only one action. Overall, we observe that on lower levels the image-only traces outperform the text-only actions, whereas on higher difficulties the text-only output outperforms. We believe that reporting both metrics offers a more comprehensive evaluation across visual and textual outputs.
>
> **[W2 – Comparison of UMM and MLLM Counterparts]** We thank the reviewer for identifying this issue and agree that the current presentation can be confusing. We revised the evaluation to ensure consistent and transparent comparisons:
> - **Consistent Gemini Baseline** We revised the results to use Gemini 2.5 Flash (text) as a MLLM baseline and Gemini 2.5 Flash Image (with image generation) as the UMM setting, clearly distinguishing the two modes and being aligned with the way we report performance of Gemini 3. When comparing the results between Gemini 2.5 Flash and Gemini 2.5 Flash Image in plot https://ibb.co/hxQHDLsw , we observe no significant performance difference (the observed differences in paper fold are due to output formatting issues from Gemini 2.5 Flash rather than a difference in reasoning capability). Therefore, all claims from the paper hold.
> - **Open-Source Validation (Emu 3.5)**  To eliminate reliance on proprietary models, we add Emu 3.5, a state-of-the-art open-source UMM. Refer to the updated Figure 12 https://ibb.co/SjdqRLs . Emu 3.5 shows the same trend: interleaving visual tokens does not reliably outperform text-only CoT.
> - **Terminology** We will refine our language to clarify that we define UMMs by their functional capability (interleaved generation and conditioning) rather than a specific architectural requirement and that for the text-only baseline for Emu 3.5 we test a UMM in a text-only mode (rather than a MLLM).
>
> **[W3 – Iterative vs. Single-Pass Oracle CoT]** We thank the reviewer for this suggestion and implemented an iterative oracle setting for Rush Hour. In this setup, the model receives the initial state, a sequence of history states (with corresponding actions) and predicts the next move given the next oracle state. Therefore, reducing the multi-image burden and being more aligned with the interleaved generation setting a model might encounter while producing the CoT itself. While per-step accuracy ranges from 30-50%, this does not translate to improved end-to-end task success as shown in plot https://ibb.co/LXf8gsMg . This suggests that the visual thought bottleneck is not (only) image quality or context length, but the model's inability to use visual states for sequential planning, reinforcing our main finding.
> We will include these results and clarify differences in evaluation setup (e.g., in this iterative setting we are comparing against ground truth vs. simulating the proposed actions resulting in differences in chance performance between these two settings).
>
> **[Q1 – Text Output for Figure 15]** As per ICML constraints, we cannot link to external text, but we will include representative full reasoning traces (text + images) for more tasks and models in the appendix of the final paper. An example is provided below from Rush Hour on level 3 (the instance is the sixth row in Figure 15) from Gemini 3 I:
>
> `Here is the step-by-step solution to free the red vehicle.\n\n### **Step 1: Move Vehicle E Backward**\nThe first step is to clear the path for the red car (R). The magenta car (E) is directly blocking R's path to the exit. By moving E backward along its axis, it will slide up and to the left until it rests against the black immovable obstacle.\n\nAs shown in the image above, car E has moved out of the way, leaving a clear path for the red car.\n\n### **Step 2: Move Vehicle R Forward**\nNow that the path is clear, the red car (R) can move forward along its axis. It will travel to the right until it reaches the exit on the edge of the grid.\n\nThe image above shows the final state where the red car has successfully reached the exit.\n\n```json\n{\"answer\": \"E backward, R forward\"}\n```"`
>
> We hope this addresses your concerns and encourages you to raise your score.

---

> > ### Author Rebuttal · Reviewer_d3cr · 2026-04-02
> >
> > Thank authors for including new experiments and analysis!
> > My concerns are all resolved, so I'll increase my score to 4.

---

### Official Review · Reviewer_k7o9 · 2026-03-13

**Soundness:** 4
**Presentation:** 4
**Significance:** 4
**Originality:** 4
**Overall Recommendation:** 4
**Confidence:** 3

**Summary:**

This paper introduces MENTISOCULI, a procedural, stratified benchmark with five multi-step visual reasoning tasks designed to test whether models can form and manipulate intermediate visual representations to aid reasoning. It evaluates four model families: text-only MLLMs, a latent imagery model fine-tuned with Mirage, UMMs that interleave generated images and text, and a video generator. The main finding is negative: explicit visual traces do not improve performance over text-only baselines. Oracle visual CoT and symbolic transcription experiments suggest the bottleneck is visual representation maintenance, not raw reasoning capacity.

**Compliance With Llm Reviewing Policy:**

Affirmed.

**Final Justification:**

I appreciate the authors for providing such a detailed response and additional experiments in such a short time. I will keep my score as weak accept, since there is concurrent work in this direction. Although there are some differences, as the authors noted, I do not see a clearly significant insight that sets this work apart from the other concurrent work. Overall, it is good, but not strong enough to raise the score to 5.

**Key Questions For Authors:**

Please see the questions in the weaknesses above.

**Limitations:**

yes

**Strengths And Weaknesses:**

Strengths

1. The oracle CoT ablation, where self-generated images are swapped for ground-truth visualizations inside a UMM's chain, cleanly separates generation errors from interpretation errors. I found this to be the most informative experiment in the paper. It shows not just that visual reasoning fails, but isolates where it fails.

2. The evaluation covers text-only MLLMs, latent imagery via Mirage, UMMs, and a video model, all scored through simulators and automatic raters. Few papers in this space test across such a range.

3. Well written and easy to follow.


Weaknesses

1. The tasks, input format, prompts, and answer format are the same for MLLMs and UMMs. The only difference is that UMMs are prompted to generate intermediate images. This means the benchmark does not actually test anything unique to UMMs. The evaluation pipeline is still "extract text answer, check correctness," and there is no way to verify whether the generated images causally influenced the model's reasoning or were produced as side effects.

2. The benchmark covers 5 puzzle-style tasks with 750 total instances, which is reasonable for controlled experiments but limited in domain breadth. Compared to MIRA's 20 task types across geometry, physics, puzzles, and causal reasoning with 546 examples, MentisOculi stays within a single class of multi-step spatial puzzles. Whether the findings about mental imagery generalize to other visual reasoning domains is an open question.

3. The interpretation error finding is tested only on Gemini 2.5 Flash Image, but the capability it measures, reading and acting on interleaved images, is shared by all multimodal models. Any MLLM that accepts multi-image input could be tested the same way. Would the same interpretation errors appear in GPT-5 or Qwen3 given oracle visuals? Without this comparison, it is hard to tell whether interpretation errors are specific to UMM architectures or a broader multimodal limitation.

---

> ### Author Rebuttal · Authors · 2026-03-31
>
> We thank the reviewer for the constructive feedback. We address your feedback regarding the causal role of images and the scope of our findings below.
>
> **[W1 – Are images causal or side effects?]** The reviewer raises a critical point: do generated images actually drive reasoning? We provide three lines of evidence suggesting that while models attempt to couple images to actions, they fail to do so functionally:
>
> - **Action–image alignment** We analyze the relationship between the number of generated images and proposed actions (App. Fig. 17–18, now expanded to Gemini 3-I and Emu 3.5 https://ibb.co/840MJ9P4 ). If images were mere side effects, we would expect a random distribution. Instead, we see a clustered trend near the 1:1 diagonal, particularly in stronger models like Gemini 3-I. This indicates the models are attempting to synchronize visual states with the action sequence.
> - **Actions from Images** Second, we evaluate the generated image sequences from all tested UMMs (Gemini 3 I, Gemini 2.5 I and Emu 3.5) in Figure https://ibb.co/LhPdS30s . For this we adjusted our video auto-rater to work with image sequences. Per adjacent images we expect only one action. Overall, we observe that on lower levels the image-only traces outperform the text-only actions, whereas on higher difficulties the text-only output outperforms. This weak coupling between the performances across evaluation modes, highlights the weak coupling of the different modalities.
> - **Oracle visual CoT** Our experiment on oracle visuals (refer to updated Figure 12 https://ibb.co/SjdqRLs ) also isolates causality by replacing self-generated images with ground-truth oracle states. If images were driving reasoning, this should significantly improve performance. However, we observe only limited gains in end-to-end success.
>
> Taken together, these results suggest that while images are sometimes coupled to reasoning, they are not (yet) reliably used as functional intermediate representations, but often behave as weak or noisy side products.
> We will incorporate these analyses more prominently in the main paper.
>
> **[W2 – Generalization]**
> We agree that MENTISOCULI focuses broadly on visual puzzle tasks, yet with a large variety of task mechanisms, e.g. folding, rotation, sliding. This is a deliberate and careful design choice: we prioritize procedural generation, verifiable intermediate states, high information density and controlled difficulty scaling, which are absolutely essential for isolating multi-step reasoning with mental imagery.
>
> Concretely, our benchmark is designed to satisfy properties that are often not jointly present in broader benchmarks such as MIRA, STARE, or Zebra-CoT:
> (i) high information density (preventing trivial text-based shortcuts),
> (ii) sequential manipulation (requiring state maintenance over multiple steps), and
> (iii) ground-truth visual trajectories (enabling fine-grained diagnosis of errors).
>
> This allows us to study a mechanistic limitation—the ability to maintain and use visual state over time.
> We agree that this precision comes at the cost of domain breadth. Accordingly, we position MENTISOCULI as a high-precision diagnostic benchmark, complementary to broader datasets. While we focus on spatial puzzles, the underlying requirement—maintaining and updating internal visual state across steps—is fundamental to many sequential visual tasks (e.g., robotics, multi-step geometry).
> We will clarify the scope of our claims.
>
> **[W3 – Interpretation Errors]** We extended our oracle visual CoT evaluation to additional MLLMs (Qwen3 and GPT-5.1) https://ibb.co/FkJS9f8H , as well as additional UMMs (Gemini 3 I and Emu 3.5) https://ibb.co/SjdqRLs .
> Similar to UMMs, oracle visuals for MLLMs are not a silver bullet: the gap to full performance is not closed, consistent with prior work [1, 2]. However, MLLMs benefit more from oracle visuals than UMMs, with the more capable GPT-5.1 showing the strongest gains. This strengthens our conclusion that UMMs are under-utilizing their own generated visual states due to interpretation errors.
>
> We thank the reviewer for proposing this experiment; it strengthens our analysis and will be included in the paper.
>
> [1]Li, L., et al. Unfolding spatial cognition: Evaluating multimodal models on visual simulations. arXiv preprint arXiv:2506.04633, 2025.
> [2] Zhou, Y.  et al. When visualizing is the first step to reasoning: MIRA, a benchmark for visual chain-of-thought. arXiv preprint arXiv:2511.02779, 2025.

---

> > ### Author Rebuttal · Reviewer_k7o9 · 2026-04-03
> >
> > Thanks for the detailed rebuttal. I appreciate the authors for providing such a detailed response and additional experiments in such a short time. I will keep my score as weak accept, since there is concurrent work in this direction. Although there are some differences, as the authors noted, I do not see a clearly significant insight that sets this work apart from the other concurrent work. Overall, it is good, but not strong enough to raise the score to 5.

---

> > > ### Author Response · Authors · 2026-04-07
> > >
> > > We thank the reviewer for a positive review. We want to take this opportunity to highlight our main insights, which position our paper beyond simply being another benchmark paper. We explicitly differentiate ourselves to other concurrent works on three main axes:
> > > **(1) breadth across paradigms:** unlike prior benchmark and method papers that focus primarily on MLLMs or a single training framework, we compare text-only MLLMs, explicit interleaved-image models, latent-token methods, and video models in one controlled setup;
> > > **(2) benchmark focus:** MENTISOCULI is purpose-built to isolate multi-step visual state maintenance under high information density, which we view as the core capability behind mental imagery; and
> > > **(3) failure-mode analysis:** beyond aggregate accuracy, our oracle-CoT and follow-up diagnostics separate generation errors, interpretation errors, and whether images are functionally used at all. Together with the symbolic reformulation result, this lets us argue that the bottleneck is not raw task solvability, but maintaining and using visual state over time.
> > >
> > > We see our contribution as complementary to concurrent work: less about whether visual traces can ever help, and more about why they still fail to provide reliable gains.

---

### Official Review · Reviewer_diwT · 2026-03-13

**Soundness:** 3
**Presentation:** 3
**Significance:** 3
**Originality:** 3
**Overall Recommendation:** 4
**Confidence:** 2

**Summary:**

This paper proposes a benchmark suite to test whether "visual thoughts" (interleaved image generation and conditioning) help multimodal models on high information density, sequential manipulation tasks. Tasks are procedurally generated with stratified difficulty and include ground-truth intermediate visual trajectories. The main empirical finding is negative: across several tasks and model families, interleaved imagery does not reliably improve success rates, and sometimes hurts.

**Compliance With Llm Reviewing Policy:**

Affirmed.

**Final Justification:**

The benchmark is well-designed: procedural generation with solver-based verification, stratified difficulty, and ground-truth visual CoT trajectories. The negative finding that interleaved imagery does not reliably improve spatial reasoning in UMMs is informative and well-supported after the rebuttal.

The follow-up addressed my two remaining points. On the latent paradigm, the authors ran text-only fine-tuning baselines with identical hyperparameters and found latent decoding does not outperform them, strengthening the negative result within the fine-tuned regime. The confound (small fine-tuned models vs. large foundation models) still limits generality. On video, Rush Hour remains the only tested task; the authors show other tasks produce near-zero accuracy with current video models. This explains the choice but confirms the evidence is preliminary.

Therefore, I will keep my recommendation as 4.

**Key Questions For Authors:**

- If you evaluate UMMs without filtering (i.e., accept their final text answer regardless of whether intermediate images were produced), do the conclusions about “UMMs underperform MLLMs” still hold?
- Did you try oracle visuals in a format that reduces multi-image burden (e.g., single concatenated strip image; or providing only the current state image iteratively)? Does performance change?
- For RUSH HOUR and SLIDING PUZZLE, what exactly is the action horizon used for chance, what is the action distribution, and how is “limited backtracking” implemented?

**Strengths And Weaknesses:**

**Strengths**
- The problem is well-motivated, with a clean framing of UMMs vs MLLMs as the key comparison.
- The benchmark is designed with ground-truth visual CoT trajectories and solver-based correctness verification.
- The paper is clearly written and easy to follow, with useful diagnostic analyses beyond the headline negative result.

**Weaknesses**

- My main concern is about the evaluation protocol in Sec. 3.1, where the paper "only evaluates samples with generated visualizations" and multiple figures show "samples omitted (no answer provided)" annotations. What is the omission rate for Gemini 2.5-I and 3-I per task and difficulty level?
- The conclusion is phrased generally (“visual thoughts do not yet benefit model reasoning”), but the experimental coverage is narrow for each paradigm. In particular, the latent-token result is confounded by fine-tuning (and a weaker base model), and the video result is confounded by the rater’s strict invalidation rules and the general brittleness of video generation.

---

> ### Author Rebuttal · Authors · 2026-03-31
>
> We thank the reviewer for the thoughtful and constructive feedback. We appreciate the recognition of our benchmark’s design and the clarity of our framing. We address the feedback regarding evaluation bias and model coverage below.
>
> **[W1 & Q1 – Omission Rate]** Thank you for highlighting the omission issue—this was indeed underreported. We report the omission rates in Table https://ibb.co/CssB0H1d
> In order to have reliable results, we adjusted the number of retries for Emu 3.5 to 15 and for Gemini 3 I to 9. We observe that omission rates for Hinge Folding and Sliding Puzzles are higher for the Gemini models. This is due to the images hitting the content filter, despite all images either being synthetic or sampled from ImageNet-1k, and upon manual inspection seem harmless.
> To ensure our UMM vs. MLLM conclusion was not biased by filtering, we ran an unfiltered evaluation (falling back to text-only if no image was produced after three attempts). As shown in the plot https://ibb.co/HDR3xMG3 , the differences in performance primarily manifest as increased variance. Therefore, our main conclusion remains unchanged: UMMs do not outperform MLLMs.
>
> **[W2 – Limited Coverage]** As outlined by reviewers xNHQ and k7o9 we already test across a wide span of model paradigms. In addition to the experiments we already ran in the paper, we strengthen our experimental results by adding more models into our scope:
>
> - **Expanded Model Set** We now include additional models across paradigms, including an open-source UMM (Emu 3.5), Gemini 3-I across all tasks, and an open-source video model (Wan 2.6) on Rush Hour.  Across all of these models, our main argument, that we see no effect of explicit visual thought on reasoning, persists. The updated plots with all models can be found here: Updated Figure 2 https://ibb.co/v4HrXBhB and Figure 3 https://ibb.co/5xrqswRq
> - **Latent reasoning comparison** We agree that the current latent-token result is confounded by fine-tuning and model strength. However, with the current state of latent reasoning models, i.e., no available foundation model, this shortcoming is inevitable. We will adjust our manuscript to highlight that the current comparison is not a true apples-to-apples comparison and that, despite fine-tuning, Mirage cannot reliably perform above chance at higher levels.
> - **Video evaluation robustness** We appreciate the concern regarding strict invalidation. We implement our rater to be lenient in the following ways, ensuring it can work with brittle videos:
>     - **Existing Leniency of the Auto-Rater** Our auto-rater is designed to be highly robust to generative artifacts. It disregards morphing, color changes, and the appearance of new objects or obstacles, ensuring these do not negatively impact the evaluation.
>     - **Movement Heuristics** Additionally, moves are only registered if an object moves a minimum number of pixels; agents are not required to strictly abide by task rules, i.e., move the vehicle until it touches another vehicle or a wall.
>     - **Updated Success Criterion** We identified a specific failure mode in videos (notably in Wan 2.6) where a trajectory was invalidated if the red car moved away after reaching the exit. We have corrected this by marking a trajectory as successful if the goal state is reached at any point, aligning the evaluation with our definition of success and chance computation. The updated plots report numbers with this new form of evaluation.
>
> We will update the paper to more clearly reflect these robust evaluation criteria.
>
> **[Q2 – Iterative Oracle Visuals]** We address this concern in response to [W3] from reviewer d3rc. We do not observe a performance difference in an iterative setting, enforcing our finding: Interpretation errors persist irrespective of context length.
>
> **[Q3 – Clarification on Chance Performance]**  We assume a random agent performing up to six actions, stopping early if the goal is reached. For Rush Hour, actions are sampled from movable cars, excluding the previously moved car where possible. In Sliding Puzzle actions are sampled uniformly from valid moves.
> This setup allows implicit backtracking, as the agent may take redundant or reversible actions. We will expand the appendix to fully specify action distributions and clarify in the main paper that “limited backtracking” refers to the option of performing more moves than the minimal, optimal solution calls for.
>
> We hope these clarifications address your concerns and strengthen the empirical support for our conclusions.

---

> > ### Author Rebuttal · Reviewer_diwT · 2026-04-04
> >
> > Thank you for the experiments. W1 (omission rate) is addressed. Adding Emu 3.5 and Gemini 3-I strengthens the UMM coverage.
> >
> > **W2 (narrow experimental coverage), latent paradigm.** Not fully addressed. The Latent Sketchpad result added for Reviewer xNHQ uses the same protocol as the existing latent baseline: task-specific fine-tuning on 200 samples, small base model. Both latent experiments share the same confound, so adding another does not help. Every latent experiment compares a weak fine-tuned model against strong foundation models with no fine-tuning. The paper claims "visual thoughts do not benefit reasoning" for the latent paradigm, but the honest framing is that latent visual reasoning remains untested under fair conditions. The revision should say this explicitly.
> >
> > **W2, video paradigm.** Only Rush Hour is tested with video models. One task is thin evidence for a general claim. Reviewer k7o9 raised the same point.
> >
> > Therefore, I would like to keep my overall recommendation at 4.

---

> > > ### Author Response · Authors · 2026-04-07
> > >
> > > We thank the reviewer for the constructive follow-up and for highlighting these important points. We address them below.
> > >
> > > **[W2 – Latent Reasoning Models]** We agree that current latent experiments are not fully comparable to foundation models, as they rely on task-specific fine-tuning of smaller base models.
> > > To better contextualize latent reasoning models performance, we add a preliminary comparison: we fine-tune the base models for LS and Mirage on text-only reasoning traces using identical hyperparameters (samples, epochs, and updates as in Stage 1 in both methods). We observe that latent decoding does not yield consistent performance improvements over text-only baselines (Table https://ibb.co/jZkz5CCq), while incurring higher training and inference cost. Therefore, strengthening our argument that visual thought does not (yet) result in better performance.
> > >
> > >
> > > **[W2 – Video Paradigm]** We initially attempted evaluation across all tasks using multiple prompting strategies. However, we consistently observed that current video models do not reliably follow task constraints or produce valid solution trajectories irrespective of prompt, model choice or task difficulty.
> > > For reference, we link to some of the most visually coherent generated videos for Level 2 from Veo 3.1 (the highest performing model in our setup) for Sliding Puzzle (https://ibb.co/qLd5kdx2), Form Board (https://ibb.co/qYPSw8h5), Hinge Folding (https://ibb.co/q3cfw7VQ) and Paper Folding (https://ibb.co/V6QxspX).
> > > In practice, this behavior leads to near-zero effective accuracy, as it is not possible to extract valid action sequences from the generated videos.
> > >
> > > Based on this, we chose to report results on Rush Hour, where video generation is most structured and thus most favorable to these models. We will clarify this evaluation choice and present these findings as preliminary evidence, rather than a general conclusion about the video paradigm.
> > >
> > >
> > > We hope that these new experiments address your feedback and encourage you to raise your score.

---

### Decision · Program_Chairs · 2026-04-30

**Decision:**

Accept (regular)

**Comment:**

All four reviewers rated the paper as Weak Accept, indicating consistent support across the panel. Reviewers agreed that the paper presents a well-designed, carefully controlled benchmark for studying visual thought in multimodal reasoning, with clear empirical findings and useful diagnostic analyses. While concerns remained about the breadth of evaluation, the authors’ rebuttal substantially clarified these limitations and appropriately scoped the claims. Overall, I recommend accept.